# Towards a Theoretical Understanding of Prompt Engineering: Tractability, Existence, and Generalization

## Abstract

Prompt engineering has rapidly become an indispensable tool for the effective utilization of large language models (LLMs), turning LLMs into task-specific experts without changing their weights. Despite its significant practical achievements, the theoretical advancement in this area is relatively limited. To enhance its understanding and interpretability, this paper addresses three fundamental questions in prompt engineering: the computational tractability of finding optimal prompts, existence conditions for the required prompts, and the generalizability of prompts. Precisely, we consider the problem of finding a prompt for a given query-answer dataset and a fixed transformer. We prove that deciding the existence of a perfect prompt is NP-complete, and computing an optimal prompt is NP-hard. Furthermore, we establish sufficient conditions for the existence of perfect prompts based on the structural properties of the dataset, which are also necessary in a certain sense. Finally, we derive a generalization bound demonstrating that the effectiveness of a prompt on the dataset extends to the whole data distribution when the dataset size significantly exceeds the prompt's length. In summary, our findings answer three crucial theoretical questions in prompt engineering, offering enhanced theoretical insights and some practical guidance.

## 1 Introduction

Prompt engineering has become a pivotal technique for the great success of large language models (LLMs), enhancing their accuracy, reasoning ability, and applicability across almost all areas. LLMs can solve various tasks by zero-shot or few-shot prompting, without modifying the weights of the pre-trained model (Radford et al., 2019; Brown et al., 2020). Chain of thoughts (CoT) prompting was introduced to improve the reasoning ability of LLMs by dividing a reasoning problem into sequential steps (Wei et al., 2022). Retrieval augmented generation (RAG) was proposed to use information retrieved from an external source as prompts (Lewis et al., 2020; Izacard and Grave, 2021; Shuster et al., 2021). The above prompts are either generated by the user or from an external knowledge base, which are fixed in a certain sense. On the other hand, prompt optimization was proposed to generate better prompts for specific tasks by optimizing the prompts or their embeddings (Shin et al., 2020; Gao et al., 2021; Li and Liang, 2021; Prasad et al., 2022; Pryzant et al., 2023; Zhang et al., 2022; Sun et al., 2022; Wen et al., 2023; Sabbatella et al., 2024).

Though the above research on prompts has concentrated on engineering techniques and empirical methods (Debnath et al., 2025), on the other hand, understanding the underlying principles is also very important, substantial efforts have been directed towards establishing a theoretical framework for prompt engineering. Wang et al. (2023); Petrov et al. (2024); Nakada et al. (2025); Hu et al. (2025) proved that transformers with prompts can arbitrarily approximate certain functions. Shen et al. (2024); Yang et al. (2024); Jeon et al. (2024); Hendel et al. (2023); Xie et al. (2021) gave theoretical results for prompts in the form of in-context learning (ICL), including convergence, representation alignment, and sample bounds. Bhargava et al. (2023) studied prompt engineering through control theory.

The aforementioned study has contributed to the theoretical comprehension of prompt engineering. Nevertheless, several important issues remain unresolved. In order to further advance the theoretical

framework of prompt engineering, this paper addresses three core questions that are yet to be fully resolved in the context of applying prompt engineering.

**Question 1. Computational tractability of prompt engineering.** What is the complexity of computing an optimal prompt for a given task?

**Question 2. Existence guarantees of prompt.** Under what conditions does a prompt exist that elicits the correct predictor?

**Question 3. Generalizability of prompt.** Can the effectiveness of prompts on a finite dataset be extended to the entire distribution?

To make the problems precise, let $S = \{(x_i, y_i)\}_{i=1}^N$ be a query-answer dataset from a specific task. Denote $\mathrm{last}(x_i)$ to be the last symbol of the sequence $x_i$. A prompt $\mathcal{P}$ is said to be perfect for $S$ and a transformer $\mathcal{F}$, if $\widehat{\mathcal{F}}(\mathcal{P} \oplus x_i) = y_i$ for all $i \in [N]$, where $\oplus$ is concatenation.

Regarding the first question, we have

**Theorem 1.1** (Informal). *The decision of the existence of perfect prompts is NPC and computing the optimal prompts is NP-hard.*

Finding the optimal prompts was widely thought to be intractable. The above theorem gives the first formal proof. The computational complexity is important because it describes an intrinsic property of the problem, which also provides us with insight into the possibilities in algorithm design.

Regarding the second question, we have

**Theorem 1.2** (Informal). *For any transformer $\mathcal{F}$ and dataset $S = \{(x_i, y_i)\}_{i=1}^N$, if $\mathrm{last}(x_i) = \mathrm{last}(x_j)$ and $y_i = y_j$ for all $i, j \in [N]$, and some extra condition is satisfied, then there exists a perfect prompt $\mathcal{P}$ for $S$. Moreover, the lack of $y_i = y_j$ and $\mathrm{last}(x_i) = \mathrm{last}(x_j)$ for $i, j \in [N]$ may result in a scenario where a perfect prompt is unattainable.*

The above result shows that if the queries have certain similarities and the target answers are consistent, then a perfect prompt exists. Also note that the sufficient condition is not a necessary condition. On the other hand, since deciding the existence of a perfect prompt is NPC by Theorem 1.1, it is unrealistic to find a necessary and sufficient condition that is also easy to compute.

Our findings differ from those in (Wang et al., 2023; Petrov et al., 2024; Nakada et al., 2025; Hu et al., 2025), where prompts and transformers are crafted to approximate functions. In contrast, our approach maintains a fixed model and modifies only the prompts. Moreover, we focus on prompts that provide the precise answer, rather than an approximation, which is crucial for tasks involving mathematical reasoning (Pérez et al., 2021; Merrill and Sabharwal, 2023; Feng et al., 2023).

Similar to learning problems, prompts are obtained using a finite dataset. A crucial question is whether the nice performance of the prompt on a finite dataset extends to new data? In other words, do prompts have generalizability? Regarding the third question, we have

**Theorem 1.3** (Informal). *Let $\mathcal{F}$ be a transformer and $S$ a dataset sampled i.i.d. according to a data distribution $\mathcal{D}$. Then, with high probability of $S$, for any prompt $\mathcal{P}$, there is:*

$$\left| \mathbb{E}_{(x,y)\sim D}[\mathbf{I}(\widehat{\mathcal{F}}(\mathcal{P} \oplus x) = y)] - \frac{1}{|S|} \sum_{(x,y)\in S} \mathbf{I}(\widehat{\mathcal{F}}(\mathcal{P} \oplus x) = y) \right| \le \overline{O}\left( \sqrt{\frac{\mathrm{len}(\mathcal{P})}{|S|}} \right).$$

The above theorem indicates that if the quantity of i.i.d. data significantly surpasses the prompt length, the prompt's performance can be generalized across the entire data distribution.

This result diverges from those in (Jeon et al., 2024; Xie et al., 2021), which computed the likelihood of the transformer's output under ICL via Bayesian methods, whereas we estimated the generalization of the prompted model from a finite dataset to the full distribution. Our conclusions align with uniform generalization bounds (Mohri et al., 2018), and the hypothesis space in our case comprises the functions parameterized by $\mathcal{P}$.

In summary, our results answer three basic questions in prompt engineering, improving the understanding and interpretability of prompts for transformers. Our main contributions are

- Existence of perfect prompts for a finite query-answer dataset is shown to be NPC and obtaining optimal prompts is shown to be NP hard.

- Sufficient conditions for the existence of perfect prompts for a finite query-answer dataset are given, which are also necessary in a certain sense.

- A generalization theorem for prompts is established, which shows that the performance of the prompt on finite datasets extends to the whole data distribution when the number of data is significantly larger than the length of the prompt.

## 2 RELATED WORK

**Prompt engineering** refers to a systematic set of methods to guide the LLMs to produce the desired output by designing, refining instructions without modifying the model. Radford et al. (2019) demonstrated that zero-shot prompting can elicit task behavior from a given transformer. Brown et al. (2020) showed that LLMs can tackle new tasks via few-shot prompting without fine-tuning. Schick and Schütze (2020) showed that prompting also worked on small LLMs. Lester et al. (2021) showed that a simple prompt is comparable to full parameter tuning. Wei et al. (2022) presented CoT prompting to enhance the reasoning capabilities of LLMs, leading to numerous powerful methods such as tree of thoughts (Yao et al., 2023) and graph of thoughts (Besta et al., 2024). RAG used external sources as prompts (Lewis et al., 2020; Izacard and Grave, 2021; Shuster et al., 2021), enabling LLMs to generate responses grounded in factual knowledge.

Prompt optimization methods were proposed to generate desired prompts. Li and Liang (2021) pointed out that prompts can be found according to gradients. Prasad et al. (2022) showed how to use gradient-free local search to generate prompts. Zhang et al. (2022) showed how to use reinforcement learning to create prompts. Sun et al. (2022) used a black-box method to find the prompts. Pryzant et al. (2023) used natural language feedback as gradients to create prompts. Wen et al. (2023) used the gradient method for the embedding layer to obtain the prompt. Refer to the recent survey for more works on prompt engineering (Debnath et al., 2025).

**Theory of prompting** was studied in several aspects. Wang et al. (2023); Petrov et al. (2024); Hu et al. (2025); Nakada et al. (2025) gave the expressive ability of transformers with prompts to approximate Lipschitz functions or certain differentiable functions. Theoretical results of prompts on ICL were given: Shen et al. (2024) gave a convergence proof of a single head, single-layer softmax Transformer for Gaussian mixture classification; Yang et al. (2024) explained ICL from the perspective of representation alignment; Hendel et al. (2023) formulated ICL as a learning problem within a hypothesis space; Jeon et al. (2024); Xie et al. (2021) used information theory to provide bounds of the Bayesian error. Bhargava et al. (2023) studied whether prompts with $\leq k$ tokens push the output distribution closer to the target distribution using control theory.

## 3 PREREQUISITE

**Notation.** In this paper, we use $O(A)$ to mean a value not greater than $cA$ for some constant $c$, and $\overline{O}$ to mean that small quantities, such as logarithms, are omitted. We use $\Omega(A)$ to mean a value not less than $cA$ for some constant $c$, and $\overline{\Omega}$ or $\overline{O}$ to mean that small quantities are omitted.

**Data.** Let $\Gamma = \{\gamma_i\}_{i=1}^{T}$ be a set of basic symbols. We assume $T \geq 3$ in this paper, which is reasonable. A sequence $(\gamma_{i_j})_{j=1}^{k}$ is called a sentence with length $k$, and $2^{\Gamma}$ is the set of all the sentences with any length. For a sentence $x$, let $\text{len}(x)$ be its length, $x[i]$ be the $i$-th symbol in $x$, $\text{last}(x) = x[\text{len}(x)]$ be the last symbol of $x$. For $x_1, x_2 \in 2^{\Gamma}$, we use $x_1 \oplus x_2$ to denote the concatenation of $x_1$ and $x_2$.

In this paper, a dataset $S$ is a finite subset of $2^{\Gamma} \times \Gamma$, that is, $S = \{(x_i, y_i)\}_{i=1}^{N}$. For a dataset $S$, denote $\text{len}(S) = \max_{(x,y) \in S}\{\text{len}(x)\}$.

**Autoregressive Transformer.** An autoregressive transformer $\mathcal{F}$ has three parts: embedding, hidden layer, and output layer.

**Embedding.** The transformer first embeds each basic symbol $\gamma_i$ into a vector $v_i \in \mathbb{R}^d$, where $v_i$ serves as an adjustable parameter in the transformer and $d$ is called the *embedding length*. Then the given sentence $x = (\gamma_{i_j})_{j=1}^n \in 2^\Gamma$ is embedded into a matrix with $n$-rows and $d$-columns: $(v_{i_1}, v_{i_2}, \ldots, v_{i_n})^\tau$, which will be the input of the first hidden layer. We do not use the encoding vector in this paper so that the transformer can handle inputs of any length.

**Hidden layer.** Firstly, we define the feedforward layer and the attention layer. For an input $x \in \mathbb{R}^{n \times d}$, the feedforward layer with width $W$ is $\text{FNN}(x) = \text{Relu}(xE_1 \oplus b)E_2$, where $E_1 \in \mathbb{R}^{d \times W}, b \in \mathbb{R}^{1 \times W}, E_2 \in \mathbb{R}^{W \times W}$ are the parameters, and $xE_1 \oplus b$ means adding the vector $b$ to each row of $xE_1$. For an input $x \in \mathbb{R}^{n \times d}$, the attention-layer with width $W$ and head $H$ is $\text{ATT}(x) = \sum_{i=1}^H \text{softmax}(xQ_iK_ix^t + M)xV_i$, where $Q_i \in \mathbb{R}^{d \times W}, K_i \in \mathbb{R}^{W \times d}, V_i \in \mathbb{R}^{d \times W}$ are parameters. $M \in \{-\infty, 0\}^{n \times n}$ is a causal mask defined as $M_{i,j} = -\infty$ if and only if $j > i$, this is the core structure of the autoregressive transformer, and a transformer without it is called non-autoregressive transformer. Let $x^i$ be the output of the $i$-th layer and $x^0$ the input. Then the $i$-th hidden layer of the transformer is

$$x^i = x^{i-1} + \text{ATT}_i(x^{i-1}) + \text{FNN}_i(x^{i-1} + \text{ATT}_i(x^{i-1})),$$

where $\text{ATT}_i$ and $\text{FNN}_i$ are the $i$-th feedforward and attention layer defined above, and the width of these layers is equal to the embedding length.

**Output layer.** The output layer performs a linear transformation for the last row of the output of the last hidden layer, written as $x_{\text{len}(x)}^L$, that is, $\mathcal{F}(x) = Wx_{\text{len}(x)}^L + b \in \mathbb{R}^T$, where $W$ and $b$ are the parameters of the output layer. Then the classification result of $\mathcal{F}(x)$, written as $\widehat{\mathcal{F}}(x)$, is $\gamma_j$, where $j = \arg\max_{i \in [T]} (\mathcal{F}(x))_i$.

Throughout the remainder of this paper, when transformers are referred to, they are specifically autoregressive transformers unless stated otherwise.

**Perfect Prompt** The commonly used way of prompt design is to add a prompt before the input query (Brown et al., 2020), that is, instead of the original input $x \in 2^\Gamma$, a prepended-prompt $\mathcal{P}$ is used to get the result $\widehat{\mathcal{F}}(\mathcal{P} \oplus x)$.

For a given specific task with queries in $2^\Gamma$ and answers in $\Gamma$, we formally define the prepended-prompt problem (PPP) as follows.

**Definition 3.1.** For a given symbol set $\Gamma$, a dataset $S = \{(x_i, y_i)\}_{i=1}^N \subset 2^\Gamma \times \Gamma$, and a transformer $\mathcal{F}$, $\text{PPP}(\Gamma, S, \mathcal{F})$ is the problem to decide whether there exists a prompt $\mathcal{P} \in 2^\Gamma$ such that $\widehat{\mathcal{F}}(\mathcal{P} \oplus x_i) = y_i$ for all $(x_i, y_i) \in S$. Furthermore, use $\text{PPP}(\Gamma, S, \mathcal{F}) = 1$ to mean that such a $\mathcal{P}$ exists, which is called a perfect prepended-prompt for $S$ and $\mathcal{F}$. Otherwise, we denote $\text{PPP}(\Gamma, S, \mathcal{F}) = 0$.

In practice, the following discrete prompt optimization (DPO) is used to find the best prepended-prompt (Shin et al., 2020; Sabbatella et al., 2024), that is, for a given dataset $S$ and $\mathcal{F}$, find a

$$\mathcal{P}^* \in \arg\max_{\mathcal{P} \in 2^\Gamma} \frac{1}{|S|} \sum_{(x,y) \in S} \mathbf{I}(\widehat{\mathcal{F}}(\mathcal{P} \oplus x) = y). \tag{1}$$

In this paper, we do not focus on specific algorithms for prompt optimization. On the other hand, we will give the computational complexity of prompt optimization.

*Remark* 3.2. Note that single-token labels are used in this paper, which is the first step in solving the three questions raised in Section 1. Also note that the single-token label case can also include many meaningful tasks, such as language recognition and other single-token classification tasks; multiple choice questions such as mathematical multiple-choice questions and disease diagnosis; single-token generation problems, such as single-token mathematical computation.

## 4 EXISTENCE CONDITIONS AND COMPUTATIONAL COMPLEXITY

In this section, we show that the decision of the existence of perfect prompts is NPC. We also give sufficient conditions for the existence of perfect prompts and conditions under which perfect prompts do not exist in some scenarios. All proofs are given in the appendix of the paper.

### 4.1 Existence of perfect prepended-prompt is NPC

First, we show that the complex structure of the transformer $\mathcal{F}$ makes it very difficult to accurately find perfect prepended-prompts.

**Theorem 4.1.** $\mathrm{PPP}(\Gamma, S, \mathcal{F})$ *is an NPC problem, based on the size of the basic symbol set* $|\Gamma|$, *the size of the dataset $S$, and the size of the parameters of $\mathcal{F}$.*

**Proof Idea.** It suffices to show that any 3-SAT problem can be reduced to this problem. Given a 3-SAT problem with $N$ variables and $M$ Bohr expressions, we can construct a symbol set $\Gamma$, a transformer $\mathcal{F}$ and a dataset $S$ with the scale $\mathrm{poly}(N, M)$ such that the existence of a perfect prepended-prompt under such $\mathcal{F}$ and $S$ is computationally equivalent to solving this 3-SAT problem.

Since the decision of perfect prepended-prompts is NPC, computing an optimal $\mathcal{P}$ is NP-hard.

**Corollary 4.2.** $\mathrm{DPO}(\Gamma, S, \mathcal{F})$ *in* (1) *is NP-hard.*

The above result shows that finding the best prepended-prompt is computationally intractable. DPO is widely thought to be intractable in the literature, but Corollary 4.2 gives the first formal proof.

### 4.2 A sufficient condition for $\mathrm{PPP}(\Gamma, S, \mathcal{F}) = 1$

As concluded in the preceding section, it is challenging to precisely decide if $\mathrm{PPP}(\Gamma, S, \mathcal{F}) = 1$. However, in this section, we will give a sufficient condition for $\mathrm{PPP}(\Gamma, S, \mathcal{F}) = 1$.

For a transformer $\mathcal{F}$, let $\mathcal{F}_{-M}$ be the non-autoregressive transformer obtained from $\mathcal{F}$ by deleting the mask $M$ in the attention layer and keeping other parameters unchanged.

**Definition 4.3.** A dataset $S = \{(x_i, y_i)\}_{i=1}^{N} \subset 2^{\Gamma} \times \Gamma$ is said to be $\mathcal{F}$-*lead* for a transformer $\mathcal{F}$, if there exists an $x_0 \in 2^{\Gamma}$ such that $\mathrm{last}(x_i) = \mathrm{last}(x_0)$ and $y_i = \widehat{\mathcal{F}}_{-M}(x_0)$ for all $i \in [N]$.

This definition requires the data $x_i$ to have some similarity (the last symbols are the same) and the target $y_i$ should be the same and in a reasonable range. Under this situation, we have

**Theorem 4.4.** *For any symbol set $\Gamma$ and transformer $\mathcal{F}$, if $S$ is an $\mathcal{F}$-lead dataset, then* $\mathrm{PPP}(\Gamma, S, \mathcal{F}) = 1$.

**Proof Idea.** We demonstrate that extending the prompt's length can enhance the prompt's effect for the output, ultimately achieving the intended outcome. Simultaneously, since all queries share the same prompt, increasing its length will inevitably lead to uniform outputs from the transformer for all queries, necessitating uniformity in the target label as well.

Theorem 4.4 verifies a common fact: prompts can produce similar results for queries with some similarity. More precisely, the transformer's outcome relies on the last line of the final hidden layer, making the concluding symbol crucial for the final result. A strongly guiding prompt tends to steer the results, often leading to similar influence for samples. By combining these two points, it can be ensured that there exist prompts that lead to the same result for samples with the same last symbol.

*Remark* 4.5. In practice, since the LLMs are very large, $\widehat{\mathcal{F}}_{-M}$ can have many possible values for queries $x$ with the given last symbol. As a consequence, perfect prompts exist for datasets with queries that have the same last symbol and the same label with a high probability. We will verify this fact with experiments in Section 6.

In Theorem 4.4, $\mathcal{F}_{-M}$ is employed, rather than $\mathcal{F}$ itself. A natural question is whether we can replace condition $y_i = \widehat{\mathcal{F}}_{-M}(x)$ by $y_i = \widehat{\mathcal{F}}(x)$? The following result shows that this is not possible.

**Proposition 4.6.** *(1) For any transformer $\mathcal{F}$ and $x \in 2^{\Gamma}$, there exist infinitely $x' \in 2^{\Gamma}$ such that* $\widehat{\mathcal{F}}_{-M}(x) = \widehat{\mathcal{F}}(x')$. *(2) For some transformer $\mathcal{F}$ and $x \in 2^{\Gamma}$, there exist only finitely $x' \in 2^{\Gamma}$ such that* $\widehat{\mathcal{F}}(x) = \widehat{\mathcal{F}}(x')$.

Result (2) in Proposition 4.6 indicates that for some $x$, if $\widehat{\mathcal{F}}(x') = \widehat{\mathcal{F}}(x)$ then $\mathrm{len}(x')$ has an upper bound. In this case, if we need to find a prompt $\mathcal{P}$ such that $\widehat{\mathcal{F}}(\mathcal{P} \oplus z) = \widehat{\mathcal{F}}(x)$ for a given query $z$, then $\mathrm{len}(\mathcal{P})$ also has an upper bound. But by result (1), $\widehat{\mathcal{F}}_{-M}$ does not have this problem, so we can continuously increase $\mathrm{len}(\mathcal{P})$ until the transformer gives the desired result, as explained in the proof

idea of Theorem 4.4, which is impossible for $\widehat{\mathcal{F}}$ due to the upper bound for $\text{len}(\mathcal{P})$. Hence, we can easily obtain the following corollary.

**Corollary 4.7.** *There exist a transformer $\mathcal{F}$ and a dataset $S = \{(x_i, y_i)\}_{i=1}^N$ such that $\text{last}(x_i) = \text{last}(x)$ and $y_i = \widehat{\mathcal{F}}(x)$ for all $i \in [N]$ and some $x \in 2^\Gamma$, but $\text{PPP}(\Gamma, S, \mathcal{F}) = 0$.*

Refer to Appendix D.3 for a detailed discussion about the relationship between $\mathcal{F}(x)$ and $\mathcal{F}_{-M}(x)$.

### 4.3 LENGTH OF PERFECT PREPENDED-PROMPT

Another key question is the length of the perfect prepended-prompt $\mathcal{P}$. It is obviously difficult to determine the shortest length of such $\mathcal{P}$. In this section, we give a rough estimation.

**Proposition 4.8.** *For any $\Gamma$ and transformer $\mathcal{F}$, there exists a constant $U_\mathcal{F}$ only depending on $\mathcal{F}$ such that, for any $\mathcal{F}$-lead dataset $S$, there exists a perfect prepended-prompt $\mathcal{P}$ satisfying $\text{len}(\mathcal{P}) \leq U_\mathcal{F} \text{len}(S)$.*

Accurately determining $U_\mathcal{F}$ is difficult, which depends on the parameters of $\mathcal{F}$. In the worst-case scenario, $U_\mathcal{F}$ could be exponential in these parameters. Furthermore, a prompt length of $\Omega(\text{len}(S))$ is necessary, as shown below.

**Proposition 4.9.** *For some $\Gamma$ and some transformer $\mathcal{F}$, there exists a constant $L_\mathcal{F}$ only depending on $\mathcal{F}$ such that, for an infinite number of $\mathcal{F}$-lead datasets $S$, if $\mathcal{P}$ is a perfect prepended-prompt of $S$ and $\mathcal{F}$, then $\text{len}(\mathcal{P}) \geq L_\mathcal{F} \text{len}(S)$.*

From Propositions 4.8 and 4.9, we have the following result:

**Corollary 4.10.** *For some fixed $\mathcal{F}$, $\text{len}(\mathcal{P}) = \Theta(\text{len}(S))$ is a necessary and sufficient condition for the length of a perfect prepended-prompt of $\mathcal{F}$-lead dataset $S$.*

*Remark* 4.11. The above result provides a theoretical basis for the widely adopted practice, that is, for a fixed $\mathcal{F}$, more detailed prompts are needed for more complicated tasks with long sentences.

Note that we have estimated the upper bound for all $\mathcal{F}$ and $\mathcal{F}$-lead dataset $S$ in Proposition 4.8, rather than calculated it precisely. The lower bound in Proposition 4.9 does not hold for any $\mathcal{F}$ and $\Gamma$, as there are indeed some nice $\mathcal{F}$ and $\Gamma$, which can make $\mathcal{P}$ shorter than $\text{len}(S)$. In practice, $F$ is fixed while $S$ is variable, so it is reasonable to find the relationship between the $S$ and prepended-prompt.

### 4.4 NECESSARY CONDITIONS ON $S$ FOR $\text{PPP}(\Gamma, S, \mathcal{F}) = 1$ TO BE VALID FOR ALL $\mathcal{F}$

In this section, we show that when the "data similarity" and "same target" conditions for $S$ are not satisfied, $\text{PPP}(\Gamma, S, \mathcal{F}) = 0$ can happen for some $\mathcal{F}$.

For a transformer $\mathcal{F}$ and $\gamma \in \Gamma$, let

$$S_{\mathcal{F},\gamma} = \{\widehat{\mathcal{F}}(x) : x \in 2^\Gamma \text{ s.t. } \text{last}(x) = \gamma\} \subset \Gamma,$$

which is the set of results of $\mathcal{F}$ for all those sentences whose last element is $\gamma$.

Then we have the following obvious necessary condition for the existence of a perfect prepended-prompt, since the last symbols of $x$ and $\mathcal{P} \oplus x$ are always the same.

**Proposition 4.12.** *Let $S = \{(x_i, y_i)\}_{i=1}^N$ and $\mathcal{F}$ be a transformer. If $y_i \notin S_{\mathcal{F},\text{last}(x_i)}$ for some $i \in [N]$, then $\text{PPP}(\Gamma, S, \mathcal{F}) = 0$.*

Proposition 4.12 provides a necessary condition for $\text{PPP}(\Gamma, S, \mathcal{F}) = 1$. Note that Proposition 4.12 is quite different from the conditions in Section 4.2 and it is not a sufficient condition, so even if $y_i \in S_{\mathcal{F},\text{last}(x_i)}$ for all $i \in [N]$, $\text{PPP}(\Gamma, S, \mathcal{F}) = 0$ may still happen, as shown below.

As mentioned in Section 4.2, prompts will produce similar results for inputs with some similarity; more precisely, that is $y_i = y_j$ and $\text{last}(x_i) = \text{last}(x_j)$ for all $i, j \in [N]$ in a dataset $S = \{(x_i, y_i)\}_{i=1}^N$. We will show that without these two conditions, perfect prepended-prompts may not exist in the following two propositions, respectively.

**Proposition 4.13** (About the target). *For any given $\Gamma$ and $S = \{(x_i, y_i)\}_{i=1}^N \subset 2^\Gamma \times \Gamma$ satisfying (1) $y_s \neq y_t$ for some $s, t \in [N]$ and (2) $\text{last}(x_i) = \text{last}(x_j)$ for all $i, j \in [N]$, there exists a transformer $\mathcal{F}$ such that (3) $y_i \in S_{\mathcal{F},\text{last}(x_i)}$ for all $i \in [N]$, but (4) $\text{PPP}(\Gamma, S, \mathcal{F}) = 0$.*

Observe that condition (2) relates to the "data similarity" condition in Definition 4.3, while condition (3) corresponds to Proposition 4.12.

**Proposition 4.14** (About the last symbol). *For any given $\Gamma$ and $S = \{(x_i, y_i)\}_{i=1}^N \subset 2^\Gamma \times \Gamma$ satisfying (1) $\mathrm{last}(x_s) \neq \mathrm{last}(x_t)$ for some $s, t \in [N]$ and (2) $y_i = y_j$ for all $i, j \in [N]$, there exists a transformer $\mathcal{F}$ such that (3) $y_i \in S_{\mathcal{F}, \mathrm{last}(x_i)}$ for all $i \in [N]$, but (4) $\mathrm{PPP}(\Gamma, S, \mathcal{F}) = 0$.*

Observe that condition (2) relates to the "same target" condition in Definition 4.3.

*Remark* 4.15. These two propositions imply that using one prompt to solve tasks with diverse queries and answers is sometimes impossible. Combined with Section 4.2, an effective way to increase the power of prompting is to split the task into sub-tasks and use different prompts for each sub-task.

*Remark* 4.16. The above propositions do not demonstrate that $\mathrm{PPP}(\Gamma, S, \mathcal{F}) = 0$ for all $\mathcal{F}$ and $S$ failing to meet the conditions in Definition 4.3; hence, these conditions are not necessary conditions on the view of $\mathcal{F}$ and $S$. They suggest that if these conditions are not satisfied for $S$, there can be some $\mathcal{F}$ satisfied $\mathrm{PPP}(\Gamma, S, \mathcal{F}) = 0$, from this view, such two conditions are the necessary conditions on $S$ for $\mathrm{PPP}(\Gamma, S, \mathcal{F}) = 1$ to be valid for all $\mathcal{F}$.

## 5 GENERALIZATION OF PREPENDED-PROMPT

We discussed the existence of perfect prompts on finite datasets in the preceding section. Another important issue with prompt engineering is whether prompts generated on finite datasets can be effectively generalized to more samples. In this section, we discuss whether prompts for an iid dataset can be generalized to the whole data distribution.

We adopt the usual setting for deriving generalization bounds (Mohri et al., 2018). Let $\mathcal{D}$ be a distribution over $2^\Gamma \times \Gamma$ and $S = \{(x_i, y_i)\}_{i=1}^N$ a dataset sampled iid according to $\mathcal{D}$. Then for a given prompt $\mathcal{P}$, the gap between the population accuracy

$$\mathcal{A}_{\mathcal{F}, \mathcal{P}, \mathcal{D}} = \mathbb{E}_{(x,y) \sim D}[\mathbf{I}(\widehat{\mathcal{F}}(\mathcal{P} \oplus x) = y)]$$

and the empirical accuracy

$$\mathcal{A}_{\mathcal{F}, \mathcal{P}, S} = \frac{1}{|S|} \sum_{(x,y) \in S} \mathbf{I}(\widehat{\mathcal{F}}(\mathcal{P} \oplus x) = y)$$

can be used to measure the generalizability of the prompt. Generally speaking, $\mathcal{D}$ represents the target distribution. If the gap between these two is small, then prompts that can achieve the target on a finite set can also achieve the desired target on the whole distribution.

We now give a uniform generalization bound for prompting.

**Theorem 5.1.** *For any $\Gamma = \{\gamma_i\}_{i=1}^T$, $\mathcal{D}$, $\mathcal{F}$, and $N, L \in \mathbb{Z}_+$, with probability $1 - \delta$ of $S \sim \mathcal{D}^N$, for any $\mathcal{P} \in 2^\Gamma$ satisfying $\mathrm{len}(\mathcal{P}) \leq L$, we have*

$$|\mathcal{A}_{\mathcal{F}, \mathcal{P}, \mathcal{D}} - \mathcal{A}_{\mathcal{F}, \mathcal{P}, S}| \leq \frac{2\sqrt{2L \ln T} + \sqrt{0.5 \ln 1/\delta}}{\sqrt{N}}.$$

**Proof Idea.** We treat the set of $\mathcal{F}(\mathcal{P} \oplus x)$ as a hypothesis space that uses the prompt $\mathcal{P}$ as the controllable parameters. Then we can use the generalization bound based on Rademacher complexity (Mohri et al., 2018) on such a hypothesis space to prove the theorem.

Based on Theorem 5.1, we have the following result about the choice of the number of data:

**Corollary 5.2.** *For a prompt $\mathcal{P} \in 2^\Gamma$ of length $L$ and $\epsilon, \delta \in (0, 1)$, if $N \geq (2\sqrt{2L \ln T} + \sqrt{0.5 \ln 1/\delta})^2/\epsilon^2$, then we have $|\mathcal{A}_{\mathcal{F}, \mathcal{P}, \mathcal{D}} - \mathcal{A}_{\mathcal{F}, \mathcal{P}, S}| \leq \epsilon$ with probability $1 - \delta$.*

*Remark* 5.3. This result provides practical guidance for generalization. If the size of the dataset is significantly larger than the length of the prompts, then performance on $S$ can be generalized to the entire distribution. Note that the above bounds do not depend on $\mathcal{D}$ or $\mathcal{F}$, because in our case, the true parameters are the prompts. Also note that the target labels in $S$ are arbitrary.

However, when the length of $\mathcal{P}$ is not limited, such generalization does not hold, as shown below.

**Proposition 5.4.** *For any* $\Gamma$, $N \in \mathbb{Z}_+$ *and* $\epsilon \in (0,1)$, *if the data distribution* $\mathcal{D}$ *satisfies that* $\max_{\{x_i\}_{i=1}^N \subset 2^\kappa} \{\sum_{i=1}^N \mathbb{P}_{x \sim D}(x = x_i)\} \leq \epsilon$, *then there exist a transformer* $\mathcal{F}$ *and a* $\mathcal{P}$ *such that with probability* $1 - \epsilon$ *of* $S \sim D^N$, *it holds* $|\mathcal{A}_{\mathcal{F},\mathcal{P},\mathcal{D}} - \mathcal{A}_{\mathcal{F},\mathcal{P},S}| \geq 1 - 2\epsilon$.

This phenomenon is similar to the "overfitting" in neural networks: when prompts are too fixated on improving accuracy on $S$, they lose generalizability.

The condition $\max_{\{x_i\}_{i=1}^N \subset 2^\kappa} \{\sum_{i=1}^N \mathbb{P}_{x \sim D}(x = x_i)\} \leq \epsilon$ means that the probability in $\mathcal{D}$ is not concentrated on $N$ points, and this condition is natural, as most distributions do not concentrate on a few points, such as the uniform distribution, normal distribution, and so on. If this requirement is not satisfied, the probability distribution $\mathcal{D}$ should be focused on a few crucial points, and these significant points are naturally included in the sample set $S$ when sampling, thereby preventing any overfitting.

On the other hand, we will show that overfitting on the prompt only happens when there is a small number $N$ of data. When fixing the distribution and $N$ is large enough, even if the length of the prompt is not limited, overfitting will not occur, as shown below.

**Proposition 5.5.** *For any given* $\mathcal{D}$, $\mathcal{F}$ *and* $\delta, \epsilon \in (0,1)$, *there exists an* $N$ *such that if* $|S| > N$ *then with probability* $\geq 1 - \delta$ *of* $S \sim \mathcal{D}^N$, *for any* $\mathcal{P} \in 2^\Gamma$, *we have* $|\mathcal{A}_{\mathcal{F},\mathcal{P},\mathcal{D}} - \mathcal{A}_{\mathcal{F},\mathcal{P},S}| \leq \epsilon$.

However, this particular $N$ relies on $\mathcal{F}$ and $\mathcal{D}$, and it is evident that certain $\mathcal{F}$ and $\mathcal{D}$ can lead to a significantly large $N$.

# 6 EXPERIMENT

In this section, we use experiments to verify the theoretical results presented in Sections 4.2 and 4.4.

To facilitate utilizing gradients for discovering prompts, we employ two open-source models: GPT2 (Radford et al., 2019) and RoBERTa (Liu et al., 2019). The method in (Wen et al., 2023) is used to approximately solve the following problem for a dataset $S$ and a transformer $\mathcal{F}$:

$$\text{DPO}_{acc} = \max_{\mathcal{P} \in 2^\Gamma} \frac{1}{|S|} \sum_{(x,y) \in S} \mathbf{I}(\widehat{\mathcal{F}}(\mathcal{P} \oplus x) = y). \tag{2}$$

The method works as follows: the gradient method is used for the embedding layer to find a "nice embedding" which is projected back to a discrete symbol set to obtain the prompt. To ensure that the algorithm can stop, the lengths of prompts are at most 64. It is clear that $\text{DPO}_{acc} = 1$ if and only if $\text{PPP}(\Gamma, S, \mathcal{F}) = 1$. In our experiments, the numbers of basic symbols for GPT2 and RoBERTa are respectively 50257 and 50265, and the sentences all have length 16.

To confirm our theoretical findings, we conduct the following three sets of experiments.

**The sufficient condition.** We use experiments to verify Theorem 4.4 and Remark 4.5. $S$ consists of 1000 randomly generated sentences that have the same last symbol and the same target label. Then solve the optimization problem (2) to find $\text{DPO}_{acc}$. Repeat this experiment 100 times and let $c_1$ be the lowest value of $\text{DPO}_{acc}$ in the 100 runs and $s_1$ the proportion of the 100 runs that make $\text{PPP}(\Gamma, S, \mathcal{F}) = 1$. The results are given in Table 1.

It is easy to see that $s_1$ is very close to 1 for the two models, which means that datasets consisting of sentences with the same last symbol and the same target label satisfy $\text{PPP}(\Gamma, S, \mathcal{F}) = 1$ with high probability. This confirms the fact that the "same last symbol and the same target label" condition is already a sufficient condition for the existence of perfect prompts with high probability. Please refer to Remark 4.5 and "future work" for more discussion.

However, the values of $s_1$ and $c_1$ are not equal to 1. There may be two reasons: (1) The gradient optimization does not find the best prompt, which is an NP-hard problem as shown in Section C. (2) The target label and the last symbol do not satisfy the conditions in Definition 4.3 for randomly generated $S$. Refer to Proposition 4.6 and Corollary 4.7 for more discussions.

**The same target label condition.** We use experiments to verify Proposition 4.13. $S$ consists of 1000 randomly generated sentences that have the same last symbol but distinct labels. We randomly

Table 1: Experimental results to verify Theorem 4.4 and Propositions 4.13 and 4.14. $c_1, c_2, c_3$ are the lowest values of $\text{DPO}_{acc}$ in the 100 runs and $s_1, s_2, s_3$ are the proportions of runs that make $\text{PPP}(\Gamma, S, \mathcal{F}) = 1$ in the three experiments, respectively.

| Models | c1 | c2 | c3 | s1 | s2 | s3 |
|---|---|---|---|---|---|---|
| GPT2 | 99.5% | 5.0% | 98.2% | 98.0% | 0.0% | 18.0% |
| RoBERTa | 98.3% | 4.1% | 74.5% | 95.0% | 0.0% | 0.0% |

generate distinct labels 100 times, and compute $\text{DPO}_{acc}$ in (2) for them. Denote the lowest value of $\text{DPO}_{acc}$ as $c_2$ and denote the proportion of the 100 runs that make $\text{PPP}(\Gamma, S, \mathcal{F}) = 1$ as $s_2$. The results are given in Table 1.

The values of $c_2$ and $s_2$ verify Proposition 4.13. $c_2$ is very low and $s_2 = 0$ for both models, which means that the existence of distinct labels in $S$ can create much difficulty in prompting the sentences to the target labels, and it is almost impossible to find a situation such that $\text{PPP}(\Gamma, S, \mathcal{F}) = 1$.

**The same last symbol condition.** We use experiments to verify Proposition 4.14. We randomly generate 1000 sentences with the same target label. Randomly generate distinct symbols and use them as the last symbols of the sentences in $S$, and then compute $\text{DPO}_{acc}$ in (2). Run the experiment for 100 times. Denote the lowest value of $\text{DPO}_{acc}$ as $c_3$ and denote the proportion of runs that make $\text{PPP}(\Gamma, S, \mathcal{F}) = 1$ as $s_3$. The results are given in Table 1.

For RoBERTa, $c_3$ is about 74%, $\text{DPO}_{acc} \leq 85\%$ in all the 100 runs, and $s_3 = 0$. For GPT2, $c_3$ is about 98%, and out of 100 runs, $\text{DPO}_{acc}$ achieves 100% for 18 times. The results mean that different last symbols in the sentences can create some difficulties to obtain the perfect prompts, from the fact that $\text{PPP}(\Gamma, S, \mathcal{F}) = 0$ in most cases. However, $c_3$ is much higher than $c_2$, which implies that the influence of "different last symbol" condition is much weaker than "different target label" condition. Also, the results are quite different for GPT2 and RoBERTa, so the results also depend on the models.

Finally, it must be pointed out that we have $s_3 > 0$ for GPT2, this is not contradictory to Propositions 4.14. Because having the same target label and the same last symbol are not necessary conditions for $\text{PPP}(\Gamma, S, \mathcal{F}) = 1$ under view of $\mathcal{F}$ and $S$, refer to Remark 4.16 for more details.

## 7 CONCLUSION

This paper aims to further establish the theoretical framework of prompt engineering by focusing on three basic questions for prompt engineering: the computational tractability, existence conditions for perfect prompts, and generalizability. The following results are established. Deciding the existence of perfect prompts is NPC and finding the optimal prompts is NP-hard; some sufficient conditions for the existence of perfect prompts are given and the lengths of prompts are estimated; a uniform generalization theorem for prompts is proved. Finally, we use some simple experimental results to confirm our theoretical results.

Our theoretical results further enhance the understanding of prompt engineering and provide a certain level of interpretability. They also provide some practical guidance to prompt engineering, such as using long prompts for long sentences (Remark 4.11), using different prompts for diverse tasks (Remarks 4.15), and the number of data for generalization (Remark 5.3).

**Limitation and future work.** There exists a gap between the sufficient condition in Section 4.2 and the conditions in Section 4.4 and it is desirable to give refined sufficient and/or necessary conditions. Single-token target labels are used in this paper, which include many useful tasks (Refer to Remark 3.2) and are the first step in studying the three questions raised in Section 1. Generalizing the results to sequence target labels is desirable. The experimental results show that the "same last symbol and the same target label" condition leads to perfect prompts with high probability, and a theoretical study of this observation is interesting. Pretended-prompts are considered in this paper. Although this is the most common case considered in the literature, it is desirable to consider other cases.

**LLM statement.** We clarify that the motivation behind our work, the theorem and its proof, experimental design and implementation, are all independent of LLMs and are entirely implemented by our team. In this study, the function of the LLMs is confined to refining the texts.

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

## A  STATEMENTS

### ETHICS STATEMENT

Our work is based on theoretical principles, mainly aimed at enhancing the understanding and interpretability of prompt engineering, and there are no ethical or moral issues involved.

### REPRODUCIBILITY STATEMENT

Our theorems have been rigorously proven, and the experiments use relatively small and open-source models, which are easy to replicate.

## B  NOTATIONS AND PRELIMINARY RESULTS

Through the appendix, we use the following notations. For an $x \in 2^\Gamma$, let $x^m = x \oplus x \oplus x \oplus \ldots x \oplus x \in 2^\Gamma$, where $\oplus$ is the concatenation operation. For an $x \in 2^\Gamma$ and a $\gamma \in \Gamma$, let $\mathrm{Num}_\gamma(x)$ be the number of $\gamma$ in $x$. For instance, $\mathrm{Num}_+((1, +, 2, *, 3, +, 4)) = 2$.

The **scaling equivariance** is defined as follows:

**Definition B.1.** (1) $x \in 2^\Gamma$ and $z \in 2^\Gamma$ are scaling equivalent if $\mathrm{Num}_\gamma(x)/\mathrm{len}(x) = \mathrm{Num}_\gamma(z)/\mathrm{len}(z)$ for any $\gamma \in \Gamma$.

(2) A transformer $\mathcal{F}$ is scaling equivariant means that: for any scaling equivalent $x, z \in 2^\Gamma$, if $x_i = z_j$, then $\mathcal{F}_{k,i}(x) = \mathcal{F}_{k,j}(z)$ for any $k$, where $\mathcal{F}_{k,i}$ means the $i$-th row of the $k$-th hidden layer of $\mathcal{F}$.

Then we have

**Lemma B.2.** *For any transformer $\mathcal{F}$, the $\mathcal{F}_{-M}$ is scaling equivariant.*

*Proof.* Firstly, we need to prove the lemma for the first hidden layer.

Let the $i$-th hidden layer of $\mathcal{F}_{-M}$ be $\mathcal{F}_i$, then $\mathcal{F}_1(x) = \mathrm{FNN}(v_x + \mathrm{ATT}(v_x)) + v_x + \mathrm{ATT}(v_x)$, where $v_x$ is the embedding matrix of $x$. Assume that $x_i = z_j$ and $x$ and $z$ are scaling equivalent. We will prove that $(\mathcal{F}_1(x))_i = (\mathcal{F}_1(z))_j$.

In the attention layer, we have that $\mathrm{ATT}(v_x) = \mathrm{softmax}(v_x Q K v_x^T) V$, so the $i$-th row of $\mathrm{ATT}(v_x)$ is $\frac{\sum_j \mathrm{Num}_{\gamma_j}(x) e^{v_{x_i} Q K v_{\gamma_j}^T} v_{\gamma_j}}{\sum_j \mathrm{Num}_{\gamma_j}(x) e^{v_{x_i} Q K v_{\gamma_j}^T}} V$, where $v_{x_i}$ is the $i$-th row of $v_x$ and $v_{\gamma_j}$ is the embedding vector of $\gamma_j$.

So if $x_i = z_j$ and $x$ and $z$ are scaling equivalent, we have that:

$$
\begin{aligned}
\mathrm{ATT}(x)_i &= \frac{\sum_j \mathrm{Num}_{\gamma_j}(x) e^{v_{x_i} Q K v_{\gamma_j}^T} v_{\gamma_j}}{\sum_j \mathrm{Num}_{\gamma_j}(x) e^{v_{x_i} Q K v_{\gamma_j}^T}} V \\
&= \frac{\sum_j \mathrm{Num}_{\gamma_j}(x)/\mathrm{len}(x) e^{v_{x_i} Q K v_{\gamma_j}^T} v_{\gamma_j}}{\sum_j \mathrm{Num}_{\gamma_j}(x)/\mathrm{len}(x) e^{v_{x_i} Q K v_{\gamma_j}^T}} V \\
&= \frac{\sum_j \mathrm{Num}_{\gamma_j}(z)/\mathrm{len}(z) e^{v_{x_i} Q K v_{\gamma_j}^T} v_{\gamma_j}}{\sum_j \mathrm{Num}_{\gamma_j}(z)/\mathrm{len}(z) e^{v_{x_i} Q K v_{\gamma_j}^T}} V \\
&= \frac{\sum_j \mathrm{Num}_{\gamma_j}(z) e^{v_{z_j} Q K v_{\gamma_j}^T} v_{\gamma_j}}{\sum_j \mathrm{Num}_{\gamma_j}(z) e^{v_{z_j} Q K v_{\gamma_j}^T}} V \\
&= \mathrm{ATT}(z)_j.
\end{aligned}
$$

So in the attention layer, the $i$-th of $\mathrm{ATT}(x)$ and $j$-th of $\mathrm{ATT}(z)$ are the same. In the other parts in $\mathcal{F}_1$, other calculations do not involve interaction between rows, so it holds $(\mathcal{F}_1(x))_i = (\mathcal{F}_1(z))_j$.

Now, we show that if the lemma stands for the $i$-th hidden layer, it also stands for the $(i + 1)$-th hidden layer, which can directly lead to the lemma.

$\mathcal{F}_{i+1}(x) = \mathrm{FNN}_i(\mathcal{F}_i(x) + \mathrm{ATT}_i(\mathcal{F}_i(x))) + \mathcal{F}_i(x) + \mathrm{ATT}_i(\mathcal{F}_i(x))$. It is easy to see that $x_j = z_k$ implies $(\mathcal{F}_i(x))_j = (\mathcal{F}_i(x))_k$ by the assumption, we will prove that $(\mathcal{F}_{i+1}(x))_j = (\mathcal{F}_{i+1}(z))_k$.

We mainly need to consider the attention layer. For the attention layer, we have $\text{ATT}_i(\mathcal{F}_i(x)) = \text{softmax}(\mathcal{F}_i(x)Q_iK_i\mathcal{F}_i(x)^T)V_i$. Then the $j$-th of $\text{ATT}_i(\mathcal{F}_i(x))$ is $\frac{\sum_p \text{Num}_{\gamma_p}(x)e^{(\mathcal{F}_i(x))_j Q_i K_i((\mathcal{F}_i(x))_{n_p})^T}(\mathcal{F}_i(x))_{n_p}}{\sum_p \text{Num}_{\gamma_p}(x)e^{(\mathcal{F}_i(x))_j Q_i K_i((\mathcal{F}_i(x))_{n_p})^T}}V_i$, where $n_p$ is the smaller number such that $x_{n_p} = \gamma_p$. Hence, similar to before, we have

$$\frac{\sum_p \text{Num}_{\gamma_p}(x)e^{(\mathcal{F}_i(x))_j Q_i K_i((\mathcal{F}_i(x))_{n_p})^T}(\mathcal{F}_i(x))_{n_p}}{\sum_p \text{Num}_{\gamma_p}(x)e^{(\mathcal{F}_i(x))_j Q_i K_i((\mathcal{F}_i(x))_{n_p})^T}} = \frac{\sum_p \text{Num}_{\gamma_p}(z)e^{(\mathcal{F}_i(z))_k Q_i K_i((\mathcal{F}_i(z))_{n'_p})^T}(\mathcal{F}_i(z))_{n'_p}}{\sum_p \text{Num}_{\gamma_p}(z)e^{(\mathcal{F}_i(z))_k Q_i K_i((\mathcal{F}_i(z))_{n'_p})^T}},$$

using the fact $(\mathcal{F}_i(x))_j = (\mathcal{F}_i(z))_k$ and $(\mathcal{F}_i(x))_{n_p} = (\mathcal{F}_i(z))_{n'_p}$, where $n'_p$ is the smaller number such that $z_{n'_p} = \gamma_p$. So the $j$-th row of $\text{ATT}_i(\mathcal{F}_i(x))$ and the $k$-th row of $\text{ATT}_i(\mathcal{F}_i(z))$ are the same. Then similar to before, we can prove the result for the $(i+1)$-th hidden layer. $\square$

## C  PROOF OF THEOREM 4.1

Firstly, for a given prompt $\mathcal{P}$, it is easy to check whether $\text{PPP}(\Gamma, S, \mathcal{F}) = 1$ in polynomial time based on the size of $\Gamma$, $S$ and parameters of $\mathcal{F}$, so PPP is an NP problem. So to prove Theorem 4.1, it suffices to show that a NPC problem (here, we use 3-SAT) can be reduced to $\text{PPP}(\Gamma, S, \mathcal{F}) = 1$.

**Definition C.1** (The 3-SAT decision problem). Let $\{z_i\}_{i=1}^N$ be Boolean variables and $\Phi = \wedge_{j=1}^M \phi_j$, where $\phi_j = \vee_{k=1,2,3} z'_{j_k}$ is a Boolean clause and $z'_{j_k} \in \{z_i\}_{i=1}^n \cup \{\neg z_i\}_{i=1}^n$. The 3-SAT problem is to decide whether there exist values for the variables under which $\Phi = 1$.

Now we prove Theorem 4.1.

*Proof.* Let $\Phi = \wedge_{j=1}^M \phi_j$ be a 3-SAT problem as defined in Definition C.1. We first construct a $\Gamma$, a dataset $S$ and a transformer $\mathcal{F}$ based on $\Phi$.

**Step 1. Construction of $\Gamma, \mathcal{F}, S$.**

The symbol set is $\Gamma = \{\gamma_i\}_{i=1}^{N+M}$ and the dataset is $S = \{((\gamma_{j+N}), \gamma_{j+N})\}_{j=1}^M \subset 2^\Gamma \times \Gamma$, where $(\gamma_{j+N})$ is the sentence with a single symbol $\gamma_{j+N}$.

The transformer is defined as follows.

(1) The embedding layer: Each $\gamma_i$ will be embedded to a $4(N+M)$-dimensional vector $v_{\gamma_i}$, such that: for the $i \in [1, N+M]$, the $(4i-3)$-th to $4i$-th weights in the $v_{\gamma_i}$ are 1, other weights are 0. For a $x \in 2^\Gamma$, let $v(x)$ be its embedding matrix.

(2) The hidden layers: In the first hidden layer $\mathcal{F}_1(x)$, the attention layer is defined as: the $Q$ and $K$ are 0, $V = I$. The feedforward layer $\text{FNN}_1(x)$ in the first layer is defined as $\text{FNN}_1(x) = 0$.

Then based on the definition of the embedding layer, the output of the first layer when input is a $x$ is $\mathcal{F}_1(x) = v(x) + \text{softmax}(0)v(x)$. Hence, its last row is $v_{x[\text{len}(x)]} + \sum_{i=1}^{\text{len}(x)} \frac{v_{x[i]}}{\text{len}(x)}$. Name it as $N(x)$, and let $N_i(x)$ be the $i$-th weight of $N(x)$. Before defining the next layer, we define 3 kinds of Relu networks $G_i(x) : \mathbb{R} \to \mathbb{R}$ at first:

$G_1(x) = \text{Relu}(6N^2(x - 1/N)) + x$; and $G_2(x) = x + \text{Relu}((N+2)(1/N - x))$, $\mathcal{G}_3(x) = \text{Relu}(-x + 1/N) + x$.

Then we define the next hidden layer, which is also the last hidden layer, let the attention layer be 0, and the FNN layer makes the output of the last hidden layer $\mathcal{F}_l(x)$ to be:

$$\begin{aligned}
\mathcal{F}_l(x) = \ & (N_4(x), \mathcal{G}_1(N_4(x)), \mathcal{G}_2(N_4(x)), \mathcal{G}_3(N_4(x)), \\
& N_8(x), \mathcal{G}_1(N_8(x)), \mathcal{G}_2(N_8(x)), \mathcal{G}_3(N_8(x)), \\
& \dots, N_{4(N+M)}(x), \mathcal{G}_1(N_{4(N+M)}(x)), \mathcal{G}_2(N_{4(N+M)}(x)), \mathcal{G}_3(N_{4(N+M)}(x))).
\end{aligned}$$

Notice that in the $N(x)$, there are $N_{4i-3}(x) = N_{4i-2}(x) = N_{4i-1}(x) = N_{4i}(x) = \frac{\text{Num}_{\gamma_i}(x)}{\text{len}(x)} + I(x[\text{len}(x)] = \gamma_i)$, so it is easy to implement the above output.

(3) The output layer: For the first $N$ elements of the transformer output, it is always 0. For $j \in [M]$, the $N + j$ element of transformer output is defined below.

(3.1) $\mathcal{G}_1(N_{4i'}(x)) + \mathcal{G}_1(N_{4j'}(x)) + \mathcal{G}_1(N_{4k'}(x)) - 3/N - 100N^2 \sum_{i \in [N+1, M+N], i \neq N+j} N_{4i}(x)$ if $\phi_j = z_{i'} \cup z_{j'} \cup z_{k'}$ for some $i', j', k' \in [N]$;

(3.2) $\mathcal{G}_1(N_{4i'}(x)) + \mathcal{G}_1(N_{4j'}(x)) - 2.1(1/N - \mathcal{G}_2(N_{4k'}(x)) + \mathcal{G}_3(N_{4k'}(x))) + 0.1/N - 100N^2 \sum_{i \in [N+1, M+N], i \neq N+j} N_{4i}(x)$ if $\phi_j = z_{i'} \cup z_{j'} \cup \neg z_{k'}$ for some $i', j', k' \in [N]$;

(3.3): $\mathcal{G}_1(N_{4i'}(x)) - 2.1(1/N - \mathcal{G}_2(N_{4j'}(x)) + \mathcal{G}_3(N_{4j'}(x))) - 2.1(1/N - \mathcal{G}_2(N_{4k'}(x)) + \mathcal{G}_3(N_{4k'}(x))) + 0.1/N - 100N^2 \sum_{i \in [N+1, M+N], i \neq N+j} N_{4i}(x)$ if $\phi_j = z_{i'} \cup \neg z_{j'} \cup \neg z_{k'}$ for some $i', j', k' \in [N]$;

(3.4) $6.2/N - 2.1(1/N - \mathcal{G}_2(N_{4i'}(x)) + \mathcal{G}_3(N_{4i'}(x))) - 2.1(1/N - \mathcal{G}_2(N_{4j'}(x)) + \mathcal{G}_3(N_{4j'}(x))) - 2.1(1/N - \mathcal{G}_2(N_{4k'}(x)) + \mathcal{G}_3(N_{4k'}(x))) - 100N^2 \sum_{i \in [N+1, M+N], i \neq N+j} N_{4i}(x)$ if $\phi_j = \neg z_{i'} \cup \neg z_{j'} \cup \neg z_{k'}$ for some $i', j', k' \in [N]$.

It can be easily verified that this transformer can indeed be realized with $O(\text{poly}(M, N))$ parameters and requires $O(\log(\text{poly}(M, N)))$ bits for every parameter value.

**Step 2. We show that if** $\mathrm{PPP}(\Gamma, S, \mathcal{F}))$ **is decidable, then the 3-SAT problem** $\Phi$ **is also decidable.**

To be convenient, write $2.1(1/N - \mathcal{G}_2(x) + \mathcal{G}_3(x)) = 2.1(1/N - (N+1)Relu(1/N - x)) = G_4(x)$. The proof for this step has three parts.

**Part 1. If** $\mathrm{PPP}(\Gamma, S, \mathcal{F}) = 1$**, then the 3-SAT problem** $\Phi$ **has a solution.**

Assume that PPP problem has a solution $\mathcal{P}$, then we can find the following solution for the 3-SAT problem $\Phi$: $z_i = 1$ if and only if $\mathrm{Num}_{\gamma_i}(\mathcal{P})/(\text{len}(\mathcal{P}) + 1) \geq 1/N$ where $i \in [N]$. We will show why such is a solution of the 3-SAT problem $\Phi$.

To show that, we only need to show that each $\phi_j = 1$ in $\Phi$. We just prove it for $\phi_1$, others are similar. For the $((\gamma_{1+N}), \gamma_{1+N})$ in the set $S$, because $\widehat{\mathcal{F}}(\mathcal{P} \oplus \gamma_{1+N}) = \gamma_{1+N}$, write $x = \mathcal{P} \oplus \gamma_{1+N}$, we have that:

(1): if $\phi_1 = z_i \cup z_j \cup z_k$ for some $i, j, k \in [N]$, consider the definition of $\mathcal{P}$ and transformer $\mathcal{F}$, there must be $\mathcal{G}_1(N_{4i}(x)) + \mathcal{G}_1(N_{4j}(x)) + \mathcal{G}_1(N_{4k}(x)) - 3/N > 0$, consider that $\mathcal{G}_1$ is increased with input and $\mathcal{G}_1(1/N) = 1/N$, so at least one of $N_{4i}(x) \geq 1/N$, $N_{4j}(x) \geq 1/N$ and $N_{4k}(x) \geq 1/N$ stands, which implies $z_i = 1$ or $z_j = 1$ or $z_k = 1$, so we have that $\phi_1 = 1$;

(2): if $\phi_1 = z_i \cup z_j \cup \neg z_k$ for some $i, j, k \in [N]$, then we have that $\mathcal{G}_1(N_{4i}(x)) + \mathcal{G}_1(N_{4j}(x)) - G_4(N_{4k}(x)) + 0.1/N > 0$, if $N_{4k}(x) \geq 1/N$, then there are $-G_4(N_{4k}(x)) + 0.1/N = -2/N$ based on the definition of $\mathcal{G}_2$ and $G_3$, which implies $\mathcal{G}_1(N_{4i}(x)) + \mathcal{G}_1(N_{4j}(x)) > 2/N$, similar to before, then at least one of $N_{4i} \geq 1/N$ and $N_{4j} \geq 1/N$ stands. So $\neg z_k = 0$ implies $z_i = 1$ or $z_j = 1$, so $\phi_1 = 1$; when $\neg z_k = 1$, there is also $\phi_1 = 1$;

(3): if $\phi_1 = z_i \cup \neg z_j \cup \neg z_k$ for some $i, j, k \in [N]$, then we have that $\mathcal{G}_1(N_{4i}(x)) - \mathcal{G}_4(N_{4j}(x)) - \mathcal{G}_4(N_{4k}(x)) + 0.1/N > 0$, if $N_{4j}(x), N_{4k}(x) \geq 1/N$, then $-\mathcal{G}_4(N_{4j}(x)) - \mathcal{G}_4(N_{4k}(x)) + 0.1/N \leq -1/N$, which implies that $N_{4i}(x) \geq 1/N$, so $\neg z_k = 0$ and $\neg z_j = 0$ implies $z_i = 1$, so $\phi_1 = 1$; when $\neg z_k = 1$ or $\neg z_j = 1$, this is also $\phi_1 = 1$;

(4): if $\phi_j = \neg z_i \cup \neg z_j \cup \neg z_k$ for some $i, j, k \in [N]$, then we have that $6.2/N - \mathcal{G}_4(N_{4i}(x)) - \mathcal{G}_4(N_{4j}(x)) - \mathcal{G}_4(N_{4k}(x)) > 0$, consider that when $N_{4i}(x) \geq 1/N$, then $2.1(1/N - \mathcal{G}_4(N_{4i}(x))) = 2.1/N$, similar for $j, k$. When $N_i(x), N_j(x), N_k(x) \geq 1/N$, hence there are $6.2/N - \mathcal{G}_4(N_{4i}(x)) - \mathcal{G}_4(N_{4j}(x)) - \mathcal{G}_4(N_{4k}(x)) < 0$, which is contradictory to $\widehat{\mathcal{F}}(x) = \gamma_{1+N}$, so at least one of $N_{4i} < 1/N$, $N_{4j} < 1/N$ and $N_{4k} < 1/N$ stand, which implies $\neg z_i = 1$, $\neg z_j = 1$ or $\neg z_k = 1$, so $\phi_1 = 1$.

So we prove the Part One.

**Part 2. If** $\mathrm{PPP}(\Gamma, S, \mathcal{F}) = 0$**, then** $\Phi$ **has no solution or it has a solution** $z_i = 1$ **for all** $i$ **or** $z_i = 0$ **for all** $i$**.**

We just need to show that, if the 3-SAT problem $\Phi$ has a solution which is not $z_i = 1$ for all $i$ or $z_i = 0$ for all $i$, then such PPP $= 1$.

We consider the $\mathcal{P} = \gamma_{i_1} \oplus \gamma_{i_1} \oplus \gamma_{i_2} \oplus \gamma_{i_2} \dots \gamma_{i_j} \oplus \gamma_{i_j}$ where $i_j$ satisfied $z_{i_j} = 1$ in the solution of the 3-SAT problem $\Phi$. We just need to show that $\widehat{\mathcal{F}}(\mathcal{P} \oplus \gamma_{1+N}) = \gamma_{1+N}$; others are similar. Write $x = \mathcal{P} \oplus \gamma_{1+N}$. Because the solution is not $z_i = 1$ for all $i$ or $z_i = 0$ for all $i$, so we know that $\frac{N_{\gamma_i}(x)}{\text{len}(x)} \geq 1/(N-0.5)$ for any $i$ such that $z_i = 1$ in the solution of $\Phi$. Then we have that:

(1): if $\phi_1 = z_i \cup z_j \cup z_k$ for some $i, j, k \in [N]$, then at least one of $N_{4i}(x) \geq 1/(N-0.5)$, $N_{4j}(x) \geq 1/(N-0.5)$, and $N_{4k}(x) \geq 1/(N-0.5)$ stands, so

$$
\begin{aligned}
& \mathcal{G}_1(N_{4i}(x)) + \mathcal{G}_1(N_{4j}(x)) + \mathcal{G}_1(N_{4k}(x)) - 3/N - 100N^2 \sum_{t \in [N+2, M+N]} N_{4t}(x) \\
=\ & \mathcal{G}_1(N_{4i}(x)) + \mathcal{G}_1(N_{4j}(x)) + \mathcal{G}_1(N_{4k}(x)) - 3/N \\
\geq\ & 3 - 3/N > 0.
\end{aligned}
$$

We use $\mathcal{G}_1(1/(N-0.5)) > 3$ and $\mathcal{G}_1(x) \geq 0$ here.

On the other hand, for $t \neq 1$, there is $-100N^2 \sum_{i \neq N+t, i \in [N+1, M+N]} N_{4i}(x) \leq -100N^2$ because $N_{4(N+1)}(x) \geq 1$, so the $N + t$ weight has a value not more than 0. Consider that the value of the first N-weight is 0, so $\widehat{\mathcal{F}}(\mathcal{P} \oplus \gamma_{1+N}) = \gamma_{1+N}$;

(2): if $\phi_1 = z_i \cup z_j \cup \neg z_k$ for some $i, j, k \in [N]$, then at least one of $N_{4i}(x) \geq 1/(N-0.5)$, $N_{4j}(x) \geq 1/(N-0.5)$ and $N_{4k}(x) = 0$ stands, so

$$
\begin{aligned}
& \mathcal{G}_1(N_{4i}(x)) + \mathcal{G}_1(N_{4j}(x)) - \mathcal{G}_4(N_{4k}(x)) + 0.1/N \\
& -100N^2 \sum_{t \in [N+2, M+N]} N_{4t}(x) \\
=\ & \mathcal{G}_1(N_{4i}(x)) + \mathcal{G}_1(N_{4j}(x)) - \mathcal{G}_4(N_{4k}(x)) + 0.1/N \\
\geq\ & \min\{2.1 + 0.1/N, 3 - 2/N\} > 0
\end{aligned}
$$

In here, we use $\mathcal{G}_1(1/(N-0.5)) > 3$ and $G_4(0) = -2.1$. Similar to before, other weights have values not more than 0, so $\widehat{\mathcal{F}}(\mathcal{P} \oplus \gamma_{1+N}) = \gamma_{1+N}$;

(3): if $\phi_1 = z_i \cup \neg z_j \cup \neg z_k$ for some $i, j, k \in [N]$, then at least one of $N_{4i} \geq 1/(N-0.5)$, $N_{4j} = 0$ and $N_{4k} = 0$ stands, so

$$
\begin{aligned}
& \mathcal{G}_1(N_{4i}(x)) - \mathcal{G}_4(N_{4j}(x)) - \mathcal{G}_4(N_{4k}(x)) + 0.1/N \\
& -100N^2 \sum_{t \in [N+2, M+N]} N_{4t}(x) \\
=\ & \mathcal{G}_1(N_{4i}(x)) - \mathcal{G}_4(N_{4j}(x)) - \mathcal{G}_4(N_{4k}(x)) + 0.1/N \\
\geq\ & \min\{3 - 4.1/N, 2.1 - 2/N\} > 0
\end{aligned}
$$

Similar to before, other weights have a value of no more than 0, so $\widehat{\mathcal{F}}(\mathcal{P} \oplus \gamma_{1+N}) = \gamma_{1+N}$;

(4): if $\phi_1 = \neg z_i \cup \neg z_j \cup \neg z_k$ for some $i, j, k \in [N]$, then we have at least one of $N_{4i} = 0$, $N_{4j} = 0$, and $N_{4k} = 0$ standing, so

$$
\begin{aligned}
& 6.2/N - \mathcal{G}_4(N_{4i}(x)) - \mathcal{G}_4(N_{4j}(x)) - \mathcal{G}_4(N_{4k}(x)) - 100N^2 \sum_{t \in [N+2, M+N]} N_{4t}(x) \\
=\ & 6.2/N - \mathcal{G}_4(N_{4i}(x)) - \mathcal{G}_4(N_{4j}(x)) - \mathcal{G}_4(N_{4k}(x)) \\
\geq\ & 2.1 + 2/N > 0
\end{aligned}
$$

Similar as before, other weights have value not more than 0, so $\widehat{\mathcal{F}}(\mathcal{P} \oplus \gamma_{1+N}) = \gamma_{1+N}$. So we prove Part Two.

**Part 3.** Now we prove the theorem. If we can solve the $PPP$ problem, then we can solve the 3-SAT existence problem as follows:

(1) Construct such transformer $\mathcal{F}$, dataset $S$ and symbols $\Gamma$ based on $\Phi$ as before;

(2) If $\text{PPP}(\Gamma, S, \mathcal{F}) = 1$, then the 3-SAT Problem $\Phi$ has a solution;

(3) If $\text{PPP}(\Gamma, S, \mathcal{F}) = 0$, then the 3-SAT $\Phi$ problem has no solution, or the solution satisfies: all of the values of $z_i$ are the same, which is easy to check in polynomial time. $\qquad\square$

## D    PROOFS OF SECTION 4.2

### D.1    SEVERAL LEMMAS

**Lemma D.1.** *For any given transformer $\mathcal{F}$, there is an $A \in \mathbb{R}_+$ which only depends on $\mathcal{F}$ such that:*

*(1) for any $x \in 2^{\Gamma}$ and $l$, $||\mathcal{F}_l(x)||_{2,\infty} \leq A$, where $\mathcal{F}_l$ means the $l$-th hidden layer.*

*(2) for any $x \in 2^{\Gamma}$ and $l$, $||\mathcal{F}_l^{-M}(x)||_{2,\infty} \leq A$, where $\mathcal{F}_l^{-M}$ means the $l$-th hidden layer of $\mathcal{F}_{-M}$.*

*Proof.* We first prove (1).

Based on the definition of embedding vector, for any sample $x$, the input of the first hidden layer is the embedding matrix $v_x$, whose $L_{2,\infty}$ norm has an upper bound $A_0$.

To prove the lemma, we just need to prove that, for the $l$-th hidden layer, if the $L_{2,\infty}$ norm of its input has an upper bound $A_{l-1}$ for any input $x$, then the $L_{2,\infty}$ norm of its output has an upper bound $A_l$ for any input $x$.

Assume that $\mathcal{F}_l(x) = \text{FNN}(\mathcal{F}_{l-1}(x) + \text{ATT}(\mathcal{F}_{l-1}(x))) + \mathcal{F}_{l-1}(x) + \text{ATT}(\mathcal{F}_{l-1}(x))$.

In the attention layer, because the $j$-th row of the attention layer is $\frac{\sum_{i=1}^{j} e^{u_{j,i}} v_i}{\sum_{i=1}^{j} e^{u_{j,i}}} V_l$, where $u_{j,i} = v_j Q_l K_l v_i^T$, where $v_i$ are the $i$-th row of $\mathcal{F}_{l-1}(x)$ and $Q_l, K_l, V_l$ are parameters in the attention layer, consider that $||\frac{\sum_{i=1}^{j} e^{u_i} v_i}{\sum_{i=1}^{j} e^{u_i}}||_2 \leq \max_{i \leq j}\{||v_i||_2\}$, so $||\text{ATT}(\mathcal{F}_{l-1}(x))||_{2,\infty} \leq ||\mathcal{F}_{l-1}(x)||_{2,\infty} ||V_l||_2$.

Consider that $FNN$ layer does not involve interaction between rows of $\mathcal{F}_{l-1}(x)$, so the $||\text{FNN}(x)||_{2,\infty} \leq ||W_l||_2 ||x||_{2,\infty} + ||b_l||_{2,\infty}$ for any $x$, where $W_l$ and $b_l$ are the parameters in $FNN$.

Combining the result of the attention layer and FNN layer, we have that $A_l \leq ||W_l||_2(A_{l-1} + A_{l-1}||V_l||_2) + (A_{l-1} + A_{l-1}||V_l||_2) + ||b_l||_{2,\infty}$, which is what we want.

Take $A = \max_l\{A_l\}$, so we prove (1) in the lemma.

We now prove (2). The proof is similar to (1), and the difference is that the $j$-th row of the attention layer is $\frac{\sum_{i=1}^{\text{len}(x)} e^{u_{j,i}} v_i}{\sum_{i=1}^{\text{len}(x)} e^{u_{j,i}}} V$, others are the same. $\square$

**Lemma D.2.** *For any sentences $x$ and $z$, if the first $n$ symbols in $x$ and $z$ are the same, then for any transformer $\mathcal{F}$ and $j \in \mathbb{Z}_+$, the first $n$ rows in the outputs of $j$-th hidden layer of $\mathcal{F}(x)$ and $\mathcal{F}(z)$ are the same.*

*Proof.* Firstly, we prove that for the $j$-th hidden layer $\mathcal{F}^j$ of $\mathcal{F}$, if $x_1$ and $z_1$ satisfy that the first $n$ rows in $x_1$ and $z_1$ are the same, then the first rows $n$ of $\mathcal{F}^j(x_1)$ and $\mathcal{F}^j(z_1)$ are the same.

Assume that $\mathcal{F}^j$ can be written as: $\mathcal{F}^j(x) = x + \sum_{k=1}^{H} \text{softmax}(xQ_kV_kx^T + M)xK_k + \text{FNN}(x + \sum_{k=1}^{H} \text{softmax}(xQ_kV_kx^T + M)xK_k)$.

In the residual layer, the calculations in this layer do not include the interactions between different rows, just do the same transformation to each row. So when $x_1$ and $z_1$ satisfy that the first $n$ rows in $x_1$ and $z_1$ are the same, $x_1 w$ and $z_1 w$ also satisfy that the first $n$ symbols are the same.

In the attention layer, considering the definition of $M$, for any $i \in \mathbb{Z}_+$, we know that the $i$-th row of $\text{softmax}(xQ_kV_kx^T + M)$ can be written as $(x_iQ_kV_kx_1^T, x_iQ_kV_kx_2^T, x_iQ_kV_kx_3^T, \ldots, x_iQ_kV_kx_i^T, 0, 0, \ldots, 0)$, where $x_i$ is the $i$-th row of $x$. So it is easy to see that, when $x_1$ and $z_1$ satisfy that the first $n$ rows in $x_1$ and $z_1$ are the same, the first $n$ rows of $\text{softmax}(x_1Q_kV_kx_1^T + M)x_1$ and $\text{softmax}(z_1Q_kV_kz_1^T + M)z_1$ are the same. Hence, because the other parts in the attention layer do not include the interactions between different rows, we have that the whole output of the attention also satisfies that the first $n$ symbols are the same when input $x_1$ and $z_1$.

By the above result, the first $n$ rows of $x_1 + \sum_{k=1}^{H} \text{softmax}(x_1Q_kV_kx_1^T + M)x_1K_k$ and $z_1 + \sum_{k=1}^{H} \text{softmax}(z_1Q_kV_kz_1^T + M)z_1K_k$ are the same. Similar to the residual layer, the first $n$ rows of $\text{FNN}(x_1 + \sum_{k=1}^{H} \text{softmax}(x_1Q_kV_kx_1^T + M)x_1K_k)$ and $\text{FNN}(z_1 + \sum_{k=1}^{H} \text{softmax}(z_1Q_kV_kz_1^T + M)z_1K_k)$ are the same. Adding them, we can get the result.

Now, we can prove the lemma. We prove the situation for the first layer. If the first $n$ symbols in $x$ and $z$ are the same, then the first $n$ rows of their embedding matrix are also the same. According to

the above result, we see that the first $n$ rows in the output of the first hidden layer are the same when input $x$ and $z$.

If the first $n$ rows in the output of the $i$-th hidden layer are the same, according to the above result and the fact that the output of the $i$-th hidden layer is the input of the $(i + 1)$-th hidden layer, then the first $n$ rows in the output of the $(i + 1)$-th hidden layer are also the same. When input $x$ and $z$ to the transformer, we have proved that $i = 1$ is valid, so the result is valid for any $i$, and we prove the lemma. $\qquad\square$

**Lemma D.3.** *For a transformer $\mathcal{F}$ and $x \in 2^{\Gamma}$, it holds $\lim_{m \to \infty} ||\mathcal{F}(x^m) - \mathcal{F}_{-M}(x^m)||_2 = 0$.*

*Proof.* Let $\mathcal{F}_{[i]}(x)$ be the $i$-th hidden layer of $\mathcal{F}(x)$, $\mathcal{F}_{[i],j}(x)$ be the $j$-th row of $\mathcal{F}_{[i]}(x)$. Similar to $\mathcal{F}_{-M}$.

If the transformer has $L$ hidden layers, to prove the lemma, we just need to show that: $\lim_{m \to \infty} ||\mathcal{F}_{[L],(m-1)\mathrm{len}(x)+j}(x^m) - \mathcal{F}_{-M,[L],(m-1)\mathrm{len}(x)+j}(x^m)||_2 = 0$ for all $j \in [\mathrm{len}(x)]$.

To prove that, we will consider the layer by layer and show the following result: if $\lim_{m \to \infty} ||\mathcal{F}_{[i],(m-1)\mathrm{len}(x)+j}(x^m) - \mathcal{F}_{-M,[i],(m-1)\mathrm{len}(x)+j}(x^m)||_2 = 0$ for all $j \in [\mathrm{len}(x)]$ stand for $i$-th hidden layer, then we can prove that $\lim_{m \to \infty} ||\mathcal{F}_{[i+1],(m-1)\mathrm{len}(x)+j}(x^m) - \mathcal{F}_{-M,[i+1],(m-1)\mathrm{len}(x)+j}(x^m)||_2 = 0$ for all $j \in [\mathrm{len}(x)]$ is stand for $(i + 1)-$th hidden layer. See $\mathcal{F}_{[0]}(x^m)$ as the input embedding matrix of the first hidden layer which is obviously satisfied assumptions, this result directly leads to the lemma.

Firstly, based on the definition of the transformer, we have $\mathcal{F}_{[i+1]}(x^m) = \mathrm{FNN}_{i+1}(\mathcal{F}_{[i]}(x^m) + \mathrm{ATT}_{i+1}(\mathcal{F}_{[i]}(x^m))) + \mathcal{F}_{[i]}(x^m) + \mathrm{ATT}_{i+1}(\mathcal{F}_{[i]}(x^m))$. For $F_{-M}$, we just need to replace $\mathrm{ATT}_{i+1}(\mathcal{F}_{[i]}(x^m))$ with $ATT_{i+1}^{-M}(\mathcal{F}_{-M,[i]}(x^m))$.

Now we prove the result in two parts: Attention layer and FNN layer.

**The gap of the attention layer.**

In this part, we consider the gap of $\mathrm{ATT}_{i+1}(\mathcal{F}_{[i]}(x^m))$ and $ATT_{i+1}^{-M}(\mathcal{F}_{-M,[i]}(x^m))$ in the last $\mathrm{len}(x)$ rows at first. We prove that $\lim_{m \to \infty} ||\mathrm{ATT}_{i+1}(\mathcal{F}_{[i]}(x^m))[(m - 1)\mathrm{len}(x) + j] - ATT_{i+1}^{-M}(\mathcal{F}_{-M,[i]}(x^m))[(m - 1)\mathrm{len}(x) + j]|| = 0$ for any $j \in [\mathrm{len}(x)]$ when $m \to \infty$.

Based on the definition of the attention layer, we have that the $(m - 1)\mathrm{len}(x) + j$ row of $\mathrm{ATT}_{i+1}(\mathcal{F}_{[i]}(x^m))$ is:

$$\frac{\sum_{k=1}^{\mathrm{len}(x)(m-1)+j} e^{u_{\mathrm{len}(x)(m-1)+j,k}} \mathcal{F}_{[i],k}(x^m)}{\sum_{k=1}^{\mathrm{len}(x)(m-1)+j} e^{u_{\mathrm{len}(x)(m-1)+j,k}}} V_{i+1}.$$

And similarly, the $(m - 1)\mathrm{len}(x) + j$ row of $ATT_{i+1}^{-M}(\mathcal{F}_{-M,[i]}(x^m))$ is:

$$\frac{\sum_{k=1}^{m\mathrm{len}(x)} e^{u'_{\mathrm{len}(x)(m-1)+j,k}} \mathcal{F}_{-M,[i],k}(x^m)}{\sum_{k=1}^{m\mathrm{len}(x)} e^{u'_{\mathrm{len}(x)(m-1)+j,k}}} V_{i+1}.$$

Here $u_{j,k} = \mathcal{F}_{[i],j}(x^m)Q_{i+1}K_{i+1}\mathcal{F}_{[i],k}(x^m)^T$ and

$$u'_{j,k} = \mathcal{F}_{-M,[i],j}(x^m)Q_{i+1}K_{i+1}\mathcal{F}_{-M,[i],k}(x^m)^T,$$

$Q_{i+1}, K_{i+1}, V_{i+1}$ are the parameters in the $i + 1$-th attention layer.

Firstly, we will show that:

$$\lim_{m \to \infty} \left| \frac{\sum_{k=1}^{\mathrm{len}(x)(m-1)+j} e^{u_{\mathrm{len}(x)(m-1)+j,k}} - \sum_{k=1}^{m\mathrm{len}(x)} e^{u'_{\mathrm{len}(x)(m-1)+j,k}}}{m} \right| = 0.$$

It is easy to see that

$$
\begin{aligned}
&\left| \frac{\sum_{k=1}^{\mathrm{len}(x)(m-1)+j} e^{u_{\mathrm{len}(x)(m-1)+j,k}} - \sum_{k=1}^{m\mathrm{len}(x)} e^{u'_{\mathrm{len}(x)(m-1)+j,k}}}{m} \right| \\
\leq\ &\left| \frac{\sum_{k=1}^{\mathrm{len}(x)(m-1)+j} e^{u'_{\mathrm{len}(x)(m-1)+j,k}} - \sum_{k=1}^{\mathrm{len}(x)(m-1)+j} e^{u_{\mathrm{len}(x)(m-1)+j,k}}}{m} \right| \\
&+ \left| \sum_{k=\mathrm{len}(x)(m-1)+j+1}^{m\mathrm{len}(x)} e^{u'_{\mathrm{len}(x)(m-1)+j,k}} / m \right|
\end{aligned}
\tag{3}
$$

We need the following facts to prove the above limitation:

(1) For any $m_1 \leq m_2$, we have $\mathcal{F}_{[i],(m_1-1)\mathrm{len}(x)+j}(x^{m_1}) = \mathcal{F}_{[i],(m_1-1)\mathrm{len}(x)+j}(x^{m_2})$ just following the Lemma D.2. And $\mathcal{F}_{-M,[i],(m_1-1)\mathrm{len}(x)+j}(x^{m_1}) = \mathcal{F}_{-M,[i],(m_1-1)\mathrm{len}(x)+j}(x^{m_2})$, just because the $(m_1-1)\mathrm{len}(x)+j$ row of $x^{m_1}$ and $x^{m_2}$ are the same, and $\mathcal{F}^{-M}$ is scaling equivariance as shown in the Lemma B.2.

(2) Based on the assumption, for any given $\epsilon$, there is a $T(\epsilon) \in \mathbb{Z}_+$ such that $||\mathcal{F}_{[i],(m-1)\mathrm{len}(x)+j}(x^m) - \mathcal{F}_{-M,[i],(m-1)\mathrm{len}(x)+j}(x^m)||_2 < \epsilon$ for all $j \in [\mathrm{len}(x)]$ and $m \geq T(\epsilon)$.

(3) By the Lemma D.1, the $||\mathcal{F}_{[i]}(x^m)||_{2,\infty}$ and $||\mathcal{F}_{-M,[i]}(x^m)||_{2,\infty}$ have the upper bound $A_1$, so $u_{j,k}$ and $u'_{j,k}$ also have an upper bound $A_2$, $||Q_{i+1}K_{i+1}\mathcal{F}_{[i],k}(x^m)^T||_2$ and $||\mathcal{F}_{-M,[i],k}(x^m)Q_{i+1}K_{i+1}||_2$ have an upper bound $A_3$, let $A = \max\{A_1, A_2, A_3, 1\}$, which only depends on $\mathcal{F}$.

Based on the fact (3), the second part in equation 3 tends to 0 when $m \to \infty$, now we consider the first part.

By fact (2), when $m > T(\epsilon)$, there are

$$
\left| \frac{\sum_{k=1}^{\mathrm{len}(x)(m-1)+j} e^{u'_{\mathrm{len}(x)(m-1)+j,k}} - \sum_{k=1}^{\mathrm{len}(x)(m-1)+j} e^{u_{\mathrm{len}(x)(m-1)+j,k}}}{m} \right|
$$
$$
\leq \quad \left| \frac{\mathrm{len}(x)T(\epsilon)e^A}{m} \right| + \left| \frac{\sum_{k=\mathrm{len}(x)T(\epsilon)+1}^{\mathrm{len}(x)(m-1)+j} e^{u'_{\mathrm{len}(x)(m-1)+j,k}} - \sum_{k=\mathrm{len}(x)T(\epsilon)+1}^{\mathrm{len}(x)(m-1)+j} e^{u_{\mathrm{len}(x)(m-1)+j,k}}}{m} \right|
$$
$$
\leq \quad \left| \frac{\mathrm{len}(x)T(\epsilon)e^A}{m} \right| + \left| \frac{e^A(e^{2A\epsilon}-1)(m\mathrm{len}(x)-T(\epsilon)\mathrm{len}(x))}{m} \right|
$$

For the second part, we use the fact (1) and fact (3), such that

$$
|u_{\mathrm{len}(x)(m-1)+j,k} - u'_{\mathrm{len}(x)(m-1)+j,k}|
$$
$$
\leq \quad ||(\mathcal{F}_{[i],\mathrm{len}(x)(m-1)+j}(x^m) - \mathcal{F}_{-M,[i],\mathrm{len}(x)(m-1)+j}(x^m))Q_{i+1}K_{i+1}\mathcal{F}_{[i],k}(x^m)^T||_2
$$
$$
+ ||\mathcal{F}_{-M,[i],\mathrm{len}(x)(m-1)+j}(x^m)Q_{i+1}K_{i+1}(\mathcal{F}_{-M,[i],k}(x^m) - \mathcal{F}_{[i],k}(x^m))^T||_2
$$
$$
\leq \quad 2\epsilon A
$$

and $|e^{u'_{\mathrm{len}(x)(m-1)+j,k}} - e^{u_{\mathrm{len}(x)(m-1)+j,k}}| = |e^{u'_{\mathrm{len}(x)(m-1)+j,k}}(1 - e^{u_{\mathrm{len}(x)(m-1)+j,k} - u'_{\mathrm{len}(x)(m-1)+j,k}})| \leq e^A(e^{2\epsilon A} - 1)$.

So that, for any $\delta$, we just need to take $\epsilon$ such that $e^A(e^{2A\epsilon} - 1)\mathrm{len}(x) < \eta/2$, then when $m \geq \frac{2\mathrm{len}(x)T(\epsilon)e^A}{\eta}$, there is $\left| \frac{\mathrm{len}(x)T(\epsilon)e^A}{m} \right| < \eta/2$ and $\left| \frac{e^A(e^{2A\epsilon}-1)(m-T(\epsilon))\mathrm{len}(x)}{m} \right| < \eta/2$, so:

$$
\left| \frac{\sum_{k=1}^{\mathrm{len}(x)(m-1)+j} e^{u'_{\mathrm{len}(x)(m-1)+j,k}} - \sum_{k=1}^{\mathrm{len}(x)(m-1)+j} e^{u_{\mathrm{len}(x)(m-1)+j,k}}}{m} \right| < \eta,
$$

which can directly lead our result.

Secondly, we can also prove that $\lim_{m\to\infty} \frac{||f||_2}{m} = 0$, where $t = \mathrm{len}(x)$ and

$$
f = \sum_{k=1}^{tm} e^{u'_{t(m-1)+j,k}} \mathcal{F}_{-M,[i],k}(x^m) - \sum_{k=1}^{t(m-1)+j} e^{u_{t(m-1)+j,k}} \mathcal{F}_{[i],k}(x^m).
$$

Just similar to the first part.

Final, we prove that

$$
\lim_{m\to\infty} ||\mathrm{ATT}_{i+1}(\mathcal{F}_{[i]}(x^m))[(m-1)\mathrm{len}(x)+j]
$$
$$
- \mathrm{ATT}_{i+1}^{-M}(\mathcal{F}_{-M,[i]}(x^m))[(m-1)\mathrm{len}(x)+j]||_2 = 0
$$

for any $j \in [\mathrm{len}(x)]$ when $m \to \infty$.

To be convenient, we write $\sum_{k=1}^{tm} e^{u'_{tm,k}} \mathcal{F}_{-M,[i],k}(x^m) = B_{m,-M}$ and $\sum_{k=1}^{tm} e^{u'_{tm,k}} = A_{m,-M}$; $\sum_{k=1}^{t(m-1)+j} e^{u_{t(m-1)+j,k}} \mathcal{F}_{[i],k}(x^m) = B_m$ and $\sum_{k=1}^{t(m-1)+j} e^{u_{t(m-1)+j,k}} = A_m$. So based on the definition of $\mathrm{ATT}_{i+1}(\mathcal{F}_{[i]}(x^m))[(m-1)\mathrm{len}(x)+j]$ and $\mathrm{ATT}_{i+1}^{-M}(\mathcal{F}_{-M,[i]}(x^m))[(m-1)\mathrm{len}(x)+j]$, we just need to prove that $\lim_{m\to\infty} ||\frac{B_m}{A_m} - \frac{B_{m,-M}}{A_{m,-M}}||_2 = 0$, we have that:

$$||\frac{B_m}{A_m} - \frac{B_{m,-M}}{A_{m,-M}}||_2$$
$$= ||\frac{B_m(A_{m,-M}-A_m)+A_m(B_{m,-M}-B_m)}{A_m A_{m,-M}}||_2$$
$$\leq ||\frac{B_m(A_{m,-M}-A_m)}{A_m A_{m,-M}}||_2 + ||\frac{B_{m,-M}-B_m}{A_{m,-M}}||_2.$$

As mentioned before, the value of transformer and hidden layer output has an upper bound and lower bound, so $\frac{m}{A_{m,-M}}$ has an upper bound and lower bound.

So there are:

$$\lim_{m\to\infty} ||\frac{B_m(A_{m,-M}-A_m)}{A_m A_{m,-M}}||_2 = \lim_{m\to\infty} ||\frac{B_m}{A_m}||_2 \frac{m}{A_{m,-M}} \frac{|A_{m,-M}-A_m|}{m} = 0,$$

use $\lim_{m\to\infty} \frac{|A_{m,-M}-A_m|}{m} = 0$ as proved above.

And

$$\lim_{m\to\infty} ||\frac{B_{m,-M}-B_m}{A_{m,-M}}||_2 = \lim_{m\to\infty} ||\frac{B_{m,-M}-B_m}{m}||_2 \frac{m}{A_{m,-M}} = 0,$$

use $\lim_{m\to\infty} \frac{||B_{m,-M}-B_m||_2}{m} = 0$ as proved above.

Combine them, we prove the result.

**The gap of the FNN layer.**

For the $FNN$ layer, easy to see that for any Relu network $FNN_r$, there are $||FNN_r(x) - FNN_r(z)||_2 \to 0$ when $||x-z||_2 \to 0$.

So consider that $\lim_{m\to\infty} ||\text{ATT}_{i+1}(\mathcal{F}_{[i]}(x^m))[(m-1)\text{len}(x)+j] - \text{ATT}_{i+1}^{-M}(\mathcal{F}_{-M,[i]}(x^m))[(m-1)\text{len}(x)+j]|| = 0$ and $\lim_{m\to\infty} ||\mathcal{F}_{[i],(m-1)\text{len}(x)+j}(x^m) - \mathcal{F}_{-M,[i],(m-1)\text{len}(x)+j}(x^m)||_2 = 0$ for all $j \in [\text{len}(x)]$, it is obvious that

$$\lim_{m\to\infty} ||\text{FNN}_{i+1}(\text{ATT}_{i+1}(\mathcal{F}_{[i]}(x^m)) + \mathcal{F}_{[i]}(x^m))[(m-1)\text{len}(x)+j]$$
$$- \text{FNN}_{i+1}(\text{ATT}_{i+1}^{-M}(\mathcal{F}_{-M,[i]}(x^m)) + \mathcal{F}_{-M,[i]}(x^m))[(m-1)\text{len}(x)+j]||_2 = 0,$$

so we prove the limitation of $FNN$ layer.

Finally, combining the gap from the attention layer and FNN layer, we prove the result. $\square$

**Lemma D.4.** *Let $x, z \in 2^\Gamma$, $x^m[-1] \in 2^\Gamma$ be the sentence such that $x^m$ without the last element. Then we have $\lim_{m\to\infty} ||\mathcal{F}(x^m[-1] \oplus z \oplus x[\text{len}(x)]) - \mathcal{F}(x^m)||_2 = 0$.*

*Proof.* Without loss of generality, let $\gamma_1$ be the last element of $x$. We assume the $l$-th hidden layer of $\mathcal{F}(x)$ is $\mathcal{F}_l(x)$, and the $\mathcal{F}_{l,k}(x)$ is the $k$-th row of $\mathcal{F}_l(x)$. We write $x^m[-1] \oplus z \oplus \gamma_1$ as $x^{m,z}$.

By the Lemma D.2, we have that the first $m\text{len}(x) - 1$ rows of $\mathcal{F}_l(x^{m,z})$ and $\mathcal{F}_l(x^m)$ are the same.

Then we will show that if $\lim_{m\to\infty} ||\mathcal{F}_{l,\text{len}(x^m)}(x^m) - \mathcal{F}_{l,\text{len}(x^{m,z})}(x^{m,z})||_2 = 0$ stand for $l$-th hidden layer, there are $\lim_{m\to\infty} ||\mathcal{F}_{l+1,\text{len}(x^m)}(x^m) - \mathcal{F}_{l+1,\text{len}(x^{m,z})}(x^{m,z})||_2 = 0$ stand for $(l+1)$-th hidden layer. See $\mathcal{F}_0(x^m)$ and $\mathcal{F}_0(x^{m,z})$ as the embedding layer which is satisfied the above assumptions, so this result directly leads to the lemma.

By the definition of transformer, we know that $\mathcal{F}_{l+1}(x^m) = \text{FNN}_{l+1}(\mathcal{F}_l(x^m) + \text{ATT}_{l+1}(\mathcal{F}_l(x^m))) + \mathcal{F}_l(x^m) + \text{ATT}_{l+1}(\mathcal{F}_l(x^m))$, similar as for $\mathcal{F}_{l+1}(x^{m,z})$.

We prove the result in two parts.

**The gap of attention layer.**

We will prove that

$$\lim_{m\to\infty} ||\text{ATT}_{l+1}(\mathcal{F}_l(x^m))[\text{len}(x^m)] - \text{ATT}_{l+1}(\mathcal{F}_l(x^{m,z}))[\text{len}(x^{m,z})]||_2 = 0,$$

$\text{ATT}(x)[i]$ means the $i$-th row of the attention layer ATT.

Firstly, we consider the attention layer, easy to see that the last row of $\text{ATT}_{l+1}(\mathcal{F}_l(x^m))$ is $\frac{\sum_{k=1}^{\text{len}(x^m)} e^{u_k} \mathcal{F}_{l,k}(x^m)}{\sum_{k=1}^{\text{len}(x^m)} e^{u_k}} V_{l+1}$, where $u_k = \mathcal{F}_{l,\text{len}(x^m)}(x^m) Q_{l+1} K_{l+1} \mathcal{F}_{l,k}(x^m)^T$.

Similarly, the last row of $\text{ATT}(\mathcal{F}_l(x^{m,z}))$ is $\frac{\sum_{k=1}^{\text{len}(x^{m,z})} e^{u'_k} \mathcal{F}_{l,k}(x^{m,z})}{\sum_{k=1}^{\text{len}(x^{m,z})} e^{u'_k}} V_{l+1}$, where $u'_k = \mathcal{F}_{l,\text{len}(x^{m,z})}(x^{m,z}) Q_{l+1} K_{l+1} \mathcal{F}_{l,k}(x^{m,z})^T$.

Then, we prove that $\lim_{m\to\infty} |\sum_{k=1}^{\text{len}(x^{m,z})} e^{u'_k} - \sum_{k=1}^{\text{len}(x^m)} e^{u_k}|/m = 0$ at first.

We need the following facts:

(1): Based on the assumption, for any given $\epsilon$ there is a $T(\epsilon)$ such that $||\mathcal{F}_{l,\text{len}(x^m)}(x^m) - \mathcal{F}_{l,\text{len}(x^{m,z})}(x^{m,z})||_2 \leq \epsilon$ when $m \geq T(\epsilon)$.

(2): By the Lemma D.1, the $||\mathcal{F}_l(x^m)||_{2,\infty}$ and $||\mathcal{F}_l(x^{m,z})||_{2,\infty}$ have the upper bound $A_1$, so $u_k$ and $u'_k$ also have an upper bound $A_2$, $||Q_{l+1} K_{l+1} \mathcal{F}_{l,k}(x^{m,z})^T||_2$ has an upper bound $A_3$, write $A = \max\{A_1, A_2, A_3, 1\}$.

Then when $m \geq T(\epsilon)$, we have that:

$$
\begin{aligned}
& |\textstyle\sum_{k=1}^{\text{len}(x^{m,z})} e^{u'_k} - \sum_{k=1}^{\text{len}(x^m)} e^{u_k}|/m \\
\leq\ & \frac{e^A (\text{len}(z)+1)}{m} + |\textstyle\sum_{k=1}^{\text{len}(x^m)-1} e^{u'_k} - \sum_{k=1}^{\text{len}(x^m)-1} e^{u_k}|/m \\
\leq\ & \frac{e^A (\text{len}(z)+1)}{m} + \frac{e^A (e^{A\epsilon}-1)(m\text{len}(x))}{m},
\end{aligned}
$$

where we use fact 1 to get $|u_k - u'_k| \leq ||\mathcal{F}_{l,\text{len}(x^m)}(x^m) - \mathcal{F}_{l,\text{len}(x^{m,z})}(x^{m,z})||_2 ||Q_{l+1} K_{l+1} \mathcal{F}_{l,k}(x^{m,z})^T||_2 \leq A\epsilon$ for any $k \leq \text{len}(x^m) - 1$. Other parts are similar as that in the proof of Lemma D.3.

So, for any $\delta > 0$, we just need to take $\epsilon$ such that $e^A (e^{A\epsilon} - 1)\text{len}(x) < \delta/2$ and $m > T(\epsilon) + \frac{2e^A (\text{len}(z)+1)}{\delta}$, then $|\sum_{k=1}^{\text{len}(x^{m,z})} e^{u'_k} - \sum_{k=1}^{\text{len}(x^m)} e^{u_k}|/m \leq \frac{e^A (\text{len}(z)+1)}{m} + \frac{e^A (e^{A\epsilon}-1)(m\text{len}(x))}{m} \leq \delta/2 + \delta/2 = \delta$. So we prove the result.

Secondly, similarly as before, we can prove that:

$$
\lim_{m\to\infty} ||\sum_{k=1}^{\text{len}(x^{m,z})} e^{u'_k} \mathcal{F}_{l,k}(x^{m,z}) - \sum_{k=1}^{\text{len}(x^m)} e^{u_k} \mathcal{F}_{l,k}(x^m)||_2 = 0.
$$

Thirdly, similar as in the proof of Lemma D.3, we can prove that

$$
\lim_{m\to\infty} ||\text{ATT}_{l+1}(\mathcal{F}_l(x^m))[\text{len}(x^m)] - \text{ATT}_{l+1}(\mathcal{F}_l(x^{m,z}))[\text{len}(x^{m,z})]||_2 = 0.
$$

**The gap of FNN layer.**

Because $\lim_{m\to\infty} ||\text{ATT}_{l+1}(\mathcal{F}_l(x^m))[\text{len}(x^m)] - \text{ATT}_{l+1}(\mathcal{F}_l(x^{m,z}))[\text{len}(x^{m,z})]||_2 = 0$ and $\lim_{m\to\infty} ||\mathcal{F}_{l,\text{len}(x^m)}(x^m) - \mathcal{F}_{l,\text{len}(x^{m,z})}(x^{m,z})||_2 = 0$, so it is obvious that

$$
\begin{aligned}
& \lim_{m\to\infty} ||\text{FNN}_{l+1}(\text{ATT}_{l+1}(\mathcal{F}_l(x^m)) + \mathcal{F}_l(x^m))[\text{len}(x^m)] \\
& -\ \text{FNN}_{l+1}(\text{ATT}_{l+1}(\mathcal{F}_l(x^{m,z})) + \mathcal{F}_l(x^{m,z}))[\text{len}(x^{m,z})]||_2 = 0,
\end{aligned}
$$

so we prove the result.

Finally, combining the gap from the attention layer and FNN layer, we prove the lemma. $\qquad\square$

### D.2 PROOF OF THEOREM 4.4

*Proof.* Let $\mathcal{F}$ be a transformer and $S$ be a $F$-lead set. We show that, for any $x_z$ such that $x_z[\text{len}(x_z)] = x_i[\text{len}(x_i)]$ and $y_i = \widehat{\mathcal{F}}_{-M}(x_z^m)$ where $(x_i, y_i) \in S$, there exists a $m$ such that $x_z^m$ is a prompt which makes $\text{PPP}(\Gamma, S, \mathcal{F}) = 1$.

According to Lemma D.3 and Lemma D.4, for any $x_i$, there is a $m_i$ such that $\widehat{\mathcal{F}}(x_z^m \oplus x_i) = \widehat{\mathcal{F}}(x_z^m)$ (Lemma D.3, and use $x_z[\text{len}(x_z)] = x_i[\text{len}(x_i)]$) and $\widehat{\mathcal{F}}(x_z^m) = \widehat{\mathcal{F}}_{-M}(x_z^m)$ (Lemma D.4) for any

$m \geq m_i$. Combining them, we have $\widehat{\mathcal{F}}(x_z^m \oplus x_i) = \widehat{\mathcal{F}}_{-M}(x_z^m) = \widehat{\mathcal{F}}_{-M}(x_z) = y_i$, using the fact that $\mathcal{F}_{-M}$ is scaling equivalence(lemma B.2).

So by taking $m = \max_{i \in [N]}\{m_i\}$, $x_z^m$ is what we want, we prove the theorem. □

### D.3 Explanation and Proof of Proposition 4.7

For convenience, we introduce the following definition.

**Definition D.5.** A dataset $S = \{(x_i, y_i)\}_{i=1}^N \subset 2^\Gamma \times \Gamma$ is said to be *strong $\mathcal{F}$-lead* for a transformer $\mathcal{F}$, if there exists an $x \in 2^\Gamma$ such that $\text{last}(x) = \text{last}(x_i)$ and $y_i = \widehat{\mathcal{F}}(x)$ for all $i \in [N]$.

Then we have that:

**Proposition D.6.** *(1) For any transformer $\mathcal{F}$, if $S$ is $\mathcal{F}$-lead, then it is also strong $\mathcal{F}$-lead; (2) For some transformer $\mathcal{F}$, $S$ is strong $\mathcal{F}$-lead, but not $\mathcal{F}$-lead.*

*Proof.* Result (1). We just need to prove that if $\widehat{\mathcal{F}}_{-M}(x) = y$, then there exists a $z \in 2^\Gamma$ such that $\text{last}(z) = \text{last}(x)$ and $\widehat{\mathcal{F}}(z) = y$.

It is easy to see that $\widehat{\mathcal{F}}_{-M}(x^m) = y$ for any $m \in \mathbb{Z}_+$, and based on the Lemma D.3, we have $\widehat{\mathcal{F}}(x^m) = \widehat{\mathcal{F}}_{-M}(x^m) = y$ when $m$ is large enough. So we just need to take $z = x^m$ for such a large $m$.

Result (2). By the Proposition 4.6, it is obvious. Note that the proof of Proposition 4.6 does not rely on that of this proposition. □

The concept of strong $\mathcal{F}$-lead encompasses a wider spectrum of $S$ than that of $\mathcal{F}$-lead. So, why do we establish that $\text{PPP}(\Gamma, S, \mathcal{F}) = 1$ for an $S$ that is a $\mathcal{F}$-lead but not strong $\mathcal{F}$-lead? This is primarily due to a fundamental reason outlined in the following proposition 4.6(repeat in the below):

**Proposition D.7.** *(1) For any transformer $\mathcal{F}$ and $x \in 2^\Gamma$, there exist infinite $x' \in 2^\Gamma$ such that $\widehat{\mathcal{F}}_{-M}(x) = \widehat{\mathcal{F}}(x')$; (2) For some transformer $\mathcal{F}$ and $x \in 2^\Gamma$, there exists only finite $x' \in 2^\Gamma$ such that $\widehat{\mathcal{F}}(x) = \widehat{\mathcal{F}}(x')$.*

By Proposition 4.6, we obtain the following result.

**Corollary D.8.** *For some symbol set $\Gamma$ and transformer $\mathcal{F}$, a strong $\mathcal{F}$-lead dataset $S$ of $\mathcal{F}$, we have that $\text{PPP}(\Gamma, S, \mathcal{F}) = 0$.*

This corollary directly proves the corollary 4.7. The proof is easy, as shown below:

*Proof.* Based on the result (2) in the proposition 4.6, suppose that for a $x \in 2^\Gamma$ such that there only exists $K$ $x' \in 2^\Gamma$ such that $\widehat{\mathcal{F}}(x) = \widehat{\mathcal{F}}(x')$. Then consider a strong $\mathcal{F}$-lead set $S = \{(x_i, y_i)\}_{i=1}^N$ where $N > K$ satisfied that $y_i = \widehat{\mathcal{F}}(x)$, then it is easy to see that $\mathcal{P}$ does not exist because $|S| = N > K$. □

Now we prove Proposition 4.6.

*Proof.* Based on the scale equivariant of $\mathcal{F}_{-M}$ and lemma D.3, (1) of Proposition 4.6 is obvious because $\widehat{\mathcal{F}}_{-M}(x) = \widehat{\mathcal{F}}_{-M}(x^m) = \widehat{\mathcal{F}}(x^m)$ for the large enough $m$. We will prove (2) of Proposition 4.6 below.

For a matrix $V$, let $V[i]$ be the $i$-th row of $V$, $V[i]_j$ be the $j$-th weight of $V[i]$.

We will give an example for a pair of $\mathcal{F}$ and $y$ such that $\widehat{\mathcal{F}}(x) = y$ can only stand for a finite number of $x \in 2^\Gamma$.

Let $y = \gamma_1$, we consider the following transformer $\mathcal{F}$:

(1) The embedding layer: embeds the $\gamma_1$ to $v_{\gamma_1} = (0, 0, 1, 0)$, and $\gamma_j$ to $v_{\gamma_j} = (0, 0, 0, 1)$ when $j > 1$. For convenience, we also write $v_{\gamma_i}$ as $v_i$.

(2) The first layer $\mathcal{F}_1$: the attention layer is that $QK = 0$ and $V \in \{0, 1\}^{4 \times 4}$ and $V[3]_1$ and $V[4]_2$ are 1, others are 0. The FNN layer is $\text{FNN}(x) = 0$.

Easy to see that: for an input $x$, the $i$-th row of the first layer output is $v_{x_i} + \sum_{j=1}^{i} \frac{v_{x_j}}{i} V \in \mathbb{R}^4_{\geq 0}$, and there are $\mathcal{F}(x)[i]_1 + \mathcal{F}(x)[i]_2 = 1$.

We also have that $|\sum_{j=1}^{i} \frac{(v_{x_j})_k}{i} - \sum_{j=1}^{i+1} \frac{(v_{x_j})_k}{i+1}| = |(\sum_{j=1}^{i} \frac{(v_{x_j} - v_{x_{i+1}})_k}{i})/(i+1)| < 1/(i+1)$ for any $k = 1, 2, 3, 4$, where $(v_{x_i})_k$ is the $k$-th weight of $v_{x_i}$, naming this result as result $(*)$.

(3) The next several layers $\mathcal{F}_s$: all their attention layers are 0, and their FNN layer makes sure that $F_s(x)[j] = (F_{0.5,1}(F_1(x)[j]_1), F_{0,0.5}(F_1(x)[j]_2), 0, 0)$ for any $j$. In here $\mathcal{F}_{0.5,1} : \mathbb{R} \to \mathbb{R}$ is a Relu network such that $\mathcal{F}_{0.5,1}(x) \geq x$ for all $x \in [0, 1]$, $\mathcal{F}_{0.5,1}(x) \geq x + 0.2$ for the $x \in [0, 1]$ satisfied $|x - 0.5| > 0.01$ and $|x - 1| > 0.01$, and $\mathcal{F}_{0.5,1}(x) = x$ when $x \in \{0.5, 1\}$; $\mathcal{F}_{0,0.5} : \mathbb{R} \to \mathbb{R}$ is a Relu network such that $\mathcal{F}_{0,0.5}(x) \geq x$ for all $x \in [0, 1]$, $\mathcal{F}_{0,0.5}(x) \geq x + 0.2$ when $x \in [0, 1]$ satisfied $|x - 0.5| > 0.01$ and $|x| > 0.01$, and $\mathcal{F}_{0,0.5}(x) = x$ when $x \in \{0.5, 0\}$. It is easy to see that such Relu networks must exist.

After such layers, we know that each row of $\mathcal{F}_s(x)$ satisfies that $||\mathcal{F}_s(x)[i]||_1 \geq 1$; moreover, $||\mathcal{F}_s(x)[i]||_1 \geq 1.2$ when $|F_1(x)[j]_1 - 0.5| > 0.01$ and $|F_1(x)[j]_1 - 1| > 0.01$, or $|F_1(x)[j]_2 - 0.5| > 0.01$ and $|F_1(x)[j]_2| > 0.01$.

(4) The next hidden layer $\mathcal{F}_l$: the attention layer is that $QK = 0$ and $V \in \{0, 1\}^{4 \times 4}$ and the $V[1]_3$ and $V[2]_4$ are 1, others are 0. The FNN layer is $\text{FNN}(x) = 0$. The last row of $\mathcal{F}_l(x)$ is $\mathcal{F}_{ll}(x)$

(5) The FNN layer in the last hidden layer and the output layer lead to the final output of the transformer: $\mathcal{F}(x) = (0.01 - |0.75 - \mathcal{F}_{ll}(x)_3| - |0.25 - \mathcal{F}_{ll}(x)_4|, 0, -0.1, \ldots, -0.1)$, where we just need to use $|x| = Relu(x) - Relu(-x)$.

Now we show such a pair of $\mathcal{F}(x)$ and $y$ is what we want.

**Part One:** It is easy to check that $\widehat{\mathcal{F}}((\gamma_1, \gamma_2)) = \gamma_1$.

**Part Two:** Assume that $x$ such that $\widehat{\mathcal{F}}(x) = \widehat{\mathcal{F}}((\gamma_1, \gamma_2)) = \gamma_1$, we show that $\text{len}(x)$ has an upper bound, which can directly lead to our result.

If $\widehat{\mathcal{F}}(x) = \gamma_1$, we have the following results:

**Result One:** There are at least 90% proportion rows $i$ of $\mathcal{F}_s(x)$ satisfying: $\mathcal{F}_s(x)[i]_1 + \mathcal{F}_s(x)[i]_2 < 1.2$.

If not, because $\mathcal{F}_{ll}(x)_3 = \sum_{i=1}^{\text{len}(x)} \mathcal{F}_s(x)[i]_1/\text{len}(x)$ and $\mathcal{F}_{ll}(x)_4 = \sum_{i=1}^{\text{len}(x)} \mathcal{F}_s(x)[i]_2/\text{len}(x)$, it is easy to calculate that $|\mathcal{F}_{ll}(x)_3 + \mathcal{F}_{ll}(x)_4| > 0.9 * 1 + 0.1 * 1.2 = 1.02$; hence, there are $0.01 - |0.75 - \mathcal{F}_{ll}(x)_3| - |0.25 - \mathcal{F}_{ll}(x)_4| < 0$, which implies that there are $\widehat{\mathcal{F}}(x) \neq \gamma_1$.

**Result Two:** There are at least $0.9 - 0.638 > 0.25$ proportion rows $i$ satisfy: $||(\mathcal{F}_1(x)[i]_1, \mathcal{F}_1(x)[i]_2) - (0.5, 0.5)||_\infty < 0.01$.

For a $i$ such that $||\mathcal{F}_s(x)[i]||_1 < 1.2$, based on the definition of $\mathcal{F}_{0.5,1}$ and $\mathcal{F}_{0,0.5}$, there must be $||(\mathcal{F}_1(x)[i]_1, \mathcal{F}_1(x)[i]_2) - (0.5, 0.5)||_\infty < 0.01$ or $||(\mathcal{F}_1(x)[i]_1, \mathcal{F}_1(x)[i]_2) - (1, 0)||_\infty < 0.01$, use the fact that $\mathcal{F}_1(x)[i]_1 + \mathcal{F}_1(x)[i]_2 = 1$. So if $\epsilon$ proportion rows of $\mathcal{F}_1(x)$ such that $||(\mathcal{F}_1(x)[i]_1, \mathcal{F}_1(x)[i]_2) - (1, 0)||_\infty < 0.01$, then there are at least $0.9 - \epsilon$ proportion rows of $\mathcal{F}_1(x)$ such that $||(\mathcal{F}_1(x)[i]_1, \mathcal{F}_1(x)[i]_2) - (0.5, 0.5)||_\infty < 0.01$(by result one).

To make sure that $\widehat{\mathcal{F}}(x) = \gamma_1$, there should be $|0.75 - \mathcal{F}_{ll}(x)_3| < 0.01$, so there are $0.99\epsilon + 0.49(0.9 - \epsilon) < 0.76$, so $\epsilon < 0.638$, which is what we want.

Similar to before, there are at least $0.9 - \frac{0.26}{0.49} > 0.25$ proportion rows $i$ that satisfy $||(\mathcal{F}_1(x)[i]_1, \mathcal{F}_1(x)[i]_2) - (1, 0)||_\infty < 0.01$.

**Result Three:** Prove the result.

Let $\text{len}(x) = N$. Consider that $i_1$ is the last row such that $||(\mathcal{F}_1(x)[i_1]_1, \mathcal{F}_1(x)[i_1]_2) - (0.5, 0.5)||_\infty < 0.01$ and $i_2$ is the last row such that $||(\mathcal{F}_1(x)[i_2]_1, \mathcal{F}_1(x)[i_2]_2) - (1, 0)||_\infty < 0.01$.

Without losing generality, let $i_1 > i_2$, then there exists a $i_2'$ which is the first row in $(i_1, i_2]$ such that $||(\mathcal{F}_1(x)[i_2']_1, \mathcal{F}_1(x)[i_2']_2) - (1, 0)||_\infty < 0.01$.

Consider that from the row $i_1$ to $i_2'$, the first weight of $\mathcal{F}_1(x)$ changes at least $0.48$, combined with result $(*)$ and the calculation of the first hidden layer, we have that $\frac{|i_1-i_2'|}{i_1+1} \geq 0.48$, so there are $|i_1 - i_2'| \geq \frac{0.48}{1/(i_1+1)} = 0.48(i_1 + 1) > 0.48 * 0.25N = 0.12N$, use $i_1 \geq 0.25N$ here(by Result two). So there are at least $0.12N - 1$ number of $i$ such that $|\mathcal{F}_s(x)[i]| > 1.2$, which is contradictory to Result One when $0.12N - 1 \geq 0.1N$. So $N$ has an upper bound. Vice versa for $i_2 > i_1$. We prove the result. $\qquad\square$

# E  PROOFS IN SECTION 4.3

## E.1  PROOF OF PROPOSITION 4.8

We show how to estimate the length.

**Lemma E.1.** *Let $x \in 2^\Gamma$ and $\mathcal{F}$ be a transformer. Then for any $z \in 2^\Gamma$, if $||\mathcal{F}(x^m[-1] \oplus z \oplus x[\text{len}(x)]) - \mathcal{F}(x^m)||_2 \leq \epsilon$, then the value of $\text{mlen}(x)$ is at most $C_\mathcal{F}\text{len}(z)/\epsilon$, where $C_\mathcal{F}$ is a constant that only depends on $\mathcal{F}$.*

*Proof.* We follow the notations in the proof of the Lemma D.4.

Assume that $\mathcal{F}$ has $L$ hidden layers. We just need to consider the last row of the last hidden layer, i.e. we just need to prove that $||\mathcal{F}_{L,\text{len}(x^m)}(x^m) - \mathcal{F}_{L,\text{len}(x^{m,z})}(x^{m,z})||_2 \leq \epsilon$ for any $m$ such that $\text{mlen}(x) \geq C_\mathcal{F}\text{len}(z)/\epsilon$ for a $C_\mathcal{F}$ dependent on $\mathcal{F}$.

Now we assume that the above result stands for any transformer $\mathcal{F}'$ with $L - 1$ hidden layers, that is, $||\mathcal{F}'_{L-1,\text{len}(x^m)}(x^m) - \mathcal{F}'_{L-1,\text{len}(x^{m,z})}(x^{m,z})||_2 < \epsilon$ for any $m$ such that $\text{mlen}(x) > \frac{C_{\mathcal{F}'}\text{len}(z)}{\epsilon}$. Then we show that the lemma is also satisfied for the transformer with $L$ hidden layers.

Easy to see that the embedding vector of $x^m$ and $x^{m,z}$ satisfied the above result, and can be seen as a 0-hidden layer transformer, so the above result can easily lead to the lemma.

It is easy to see that $\mathcal{F}_L(x^m) = \text{FNN}_L(\text{ATT}_L(\mathcal{F}_{L-1}(x^m)) + \mathcal{F}_{L-1}(x^m)) + \text{ATT}_L(\mathcal{F}_{L-1}(x^m)) + \mathcal{F}_{L-1}(x^m)$, similar to $x^{m,z}$. Then we show such a result in two parts.

**The Attention Layer.** We show that there is a $C_\mathcal{F}^A$ dependent on $\mathcal{F}$ such that when $\text{mlen}(x) \geq \frac{C_\mathcal{F}^A\text{len}(z)}{\epsilon}$, there is $||\text{ATT}_L(\mathcal{F}_{L-1}(x^m))[\text{len}(x^m)] - \text{ATT}_L(\mathcal{F}_{L-1}(x^{m,z}))[\text{len}(x^{m,z})]||_2 \leq \epsilon$.

About the attention layer, we have that:

$$\text{ATT}_L(\mathcal{F}_{L-1}(x^m))[\text{len}(x^m)] = \frac{\sum_{k=1}^{\text{len}(x^m)} e^{u_k}\mathcal{F}_{L-1,k}(x^m)}{\sum_{k=1}^{\text{len}(x^m)} e^{u_k}}V,$$

and

$$\text{ATT}_L(\mathcal{F}_{L-1}(x^{m,z}))[\text{len}(x^{m,z})] = \frac{\sum_{k=1}^{\text{len}(x^{m,z})} e^{u_k'}\mathcal{F}_{L-1,k}(x^{m,z})}{\sum_{k=1}^{\text{len}(x^{m,z})} e^{u_k'}}V.$$

The $u_k$ and $u_k'$ are the same meaning with that in the proof of Lemma D.4. To be convenient, let $\sum_{k=1}^{\text{len}(x^{m,z})} e^{u_k'}\mathcal{F}_{L-1,k}(x^{m,z}) = (\sum_{k=1}^{\text{len}(x^m)-1} e^{u_k'}\mathcal{F}_{L-1,k}(x^{m,z}) + e^{u_{\text{len}(x^{m,z})}'}\mathcal{F}_{L-1,\text{len}(x^{m,z})}(x^{m,z})) + (\sum_{k=\text{len}(x^m)}^{\text{len}(x^{m,z})-1} e^{u_k'}\mathcal{F}_{L-1,k}(x^{m,z})) = (V_1) + (V_2)$, and $\sum_{k=1}^{\text{len}(x^{m,z})} e^{u_k'} = (\sum_{k=1}^{\text{len}(x^m)-1} e^{u_k'} + e^{u_{\text{len}(x^{m,z})}'}) + (\sum_{k=\text{len}(x^m)}^{\text{len}(x^{m,z})-1} e^{u_k'}) = (W_1) + (W_2)$. And write $\text{ATT}_L(\mathcal{F}_{L-1}(x^m))[\text{len}(x^m)] = Q/P$.

Then, we have that

$$\text{ATT}_L(\mathcal{F}_{L-1}(x^m))[\text{len}(x^m)] - \text{ATT}_L(\mathcal{F}_{L-1,\text{len}(x^{m,z})}(x^{m,z}))[\text{len}(x^{m,z})]$$
$$= \frac{(PV_1 - QW_1) + (PV_2 - QW_2)}{(W_1 + W_2)P}V.$$

And we have the following facts:

(1) Let the subset transformer of $\mathcal{F}$ consist of the first $L-1$ layer name $\mathcal{F}'$, based on the assumption, there exist a $C_{\mathcal{F}'}$ such that for any $\epsilon$, when $\mathrm{mlen}(x) \geq \frac{C_{\mathcal{F}'}\mathrm{len}(z)}{\epsilon}$, there are $||\mathcal{F}_{L-1,\mathrm{len}(x^m)}(x^m) - \mathcal{F}_{L-1,\mathrm{len}(x^{m,z})}(x^{m,z})||_2 \leq \epsilon$;

(2) By the Lemma D.1, the $||\mathcal{F}_{L-1}(x^m)||_{2,\infty}$ and $||\mathcal{F}_{L-1}(x^{m,z})||_{2,\infty}$ have the upper bound $A_1$, so $u_k$ and $u'_k$ also have an upper bound $A_2$, $||\mathcal{F}_{L-1,\mathrm{len}(x^{m,z})}(x^{m,z})QK||_2$ and $||\mathcal{F}_{L-1,\mathrm{len}(x^m)}(x^m)QK||_2$ have an upper bound $A_3$, write $A = \max\{A_1, A_2, A_3, 1\}$, which is only depended on $\mathcal{F}$.

Then when $\mathrm{mlen}(x) \geq \frac{C_{\mathcal{F}'}\mathrm{len}(z)}{\epsilon}$ and $2A\epsilon < 1$, simialr as that in the proof of the lemma D.4, we have that:

$$
\begin{aligned}
& ||V_1 - Q||_2 \\
= \ & ||(\sum_{k=1}^{\mathrm{len}(x^m)-1} e^{u'_k}\mathcal{F}_{L-1,k}(x^{m,z})) + (e^{u'_{\mathrm{len}(x^{m,z})}}\mathcal{F}_{L-1,\mathrm{len}(x^{m,z})}(x^{m,z})) \\
& - \sum_{k=1}^{\mathrm{len}(x^m)} e^{u_k}\mathcal{F}_{L-1,k}(x^m)||_2 \\
= \ & ||(\sum_{k=1}^{\mathrm{len}(x^m)-1} (e^{u'_k} - e^{u_k})\mathcal{F}_{L-1,k}(x^{m,z})) + (e^{u'_{\mathrm{len}(x^{m,z})}}\mathcal{F}_{L-1,\mathrm{len}(x^{m,z})}(x^{m,z}) \\
& - e^{u_{\mathrm{len}(x^m)}}\mathcal{F}_{L-1,\mathrm{len}(x^m)}(x^m))||_2 \\
\leq \ & (\mathrm{len}(x^m)-1)e^A(e^{A\epsilon}-1)A + ||(e^{u'_{\mathrm{len}(x^{m,z})}} - e^{u_{\mathrm{len}(x^m)}})\mathcal{F}_{L-1,\mathrm{len}(x^{m,z})}(x^{m,z})||_2 \\
& + ||e^{u_{\mathrm{len}(x^m)}}(\mathcal{F}_{L-1,\mathrm{len}(x^m)}(x^m) - \mathcal{F}_{L-1,\mathrm{len}(x^{m,z})}(x^{m,z}))||_2 \\
\leq \ & (\mathrm{len}(x^m)-1)e^A(e^{A\epsilon}-1)A + e^A(e^{2A\epsilon}-1)A + e^A\epsilon.
\end{aligned}
$$

The first equation uses the fact that the first $m\mathrm{len}(x) - 1$ rows of $\mathcal{F}_l(x^{m,z})$ and $\mathcal{F}_l(x^m)$ are the same, by the Lemma D.2. In the above second inequality sign, we use

$$
\begin{aligned}
& |u'_{\mathrm{len}(x^{m,z})} - u_{\mathrm{len}(x^m)}| \\
\leq \ & |\mathcal{F}_{L-1,\mathrm{len}(x^{m,z})}(x^{m,z})QK(\mathcal{F}_{L-1,\mathrm{len}(x^{m,z})}(x^{m,z}) - \mathcal{F}_{L-1,\mathrm{len}(x^m)}(x^m))^T| \\
& + |(\mathcal{F}_{L-1,\mathrm{len}(x^{m,z})}(x^{m,z}) - \mathcal{F}_{L-1,\mathrm{len}(x^m)}(x^m))QK\mathcal{F}_{L-1,\mathrm{len}(x^m)}(x^m)^T| \\
\leq \ & 2\epsilon A.
\end{aligned}
$$

So $||V_1 - Q||_2 \leq (\mathrm{len}(x^m)-1)e^A(e^{A\epsilon}-1)A + e^A(e^{2A\epsilon}-1)A + e^A\epsilon \leq (2A^2(\mathrm{len}(x^m)+1)+1)\epsilon e^A \leq 5A^2\mathrm{len}(x^m)\epsilon e^A = C_1\mathrm{len}(x^m)\epsilon$, when $2A\epsilon \leq 1$.

Hence, we also have

$$
\begin{aligned}
& |W_1 - P| \\
= \ & |(\sum_{k=1}^{\mathrm{len}(x^m)-1} e^{u'_k}) + (e^{u'_{\mathrm{len}(x^{m,z})}}) - \sum_{k=1}^{\mathrm{len}(x^m)} e^{u_k}| \\
= \ & |(\sum_{k=1}^{\mathrm{len}(x^m)-1} (e^{u'_k} - e^{u_k}) + (e^{u'_{\mathrm{len}(x^{m,z})}} - e^{u_{\mathrm{len}(x^m)}}))| \\
\leq \ & (\mathrm{len}(x^m)-1)e^A(e^{A\epsilon}-1) + e^A(e^{2A\epsilon}-1).
\end{aligned}
$$

So when $2\epsilon A \leq 1$, there are $|W_1 - P| \leq (\mathrm{len}(x^m)-1)e^A(e^{A\epsilon}-1) + e^A(e^{2A\epsilon}-1) \leq 2(\mathrm{len}(x^m)+1)e^A A\epsilon \leq 4\mathrm{len}(x^m)e^A A\epsilon = C_2\epsilon\mathrm{len}(x^m)$. Combine the above result, when $2\epsilon A \leq 1$, we have that:

$$
\begin{aligned}
& ||\frac{(PV_1 - QW_1) + (PV_2 - QW_2)}{(W_1 + W_2)P}||_2 \\
\leq \ & ||\frac{P(V_1 - Q)}{(W_1 + W_2)P}||_2 + ||\frac{Q(W_1 - P)}{(W_1 + W_2)P}||_2 + ||\frac{PV_2 - QW_2}{(W_1 + W_2)P}||_2 \\
\leq \ & ||\frac{C_1\mathrm{len}(x^m)\epsilon}{W_1 + W_2}||_2 + ||\frac{C_2\mathrm{len}(x^m)\epsilon Q}{(W_1 + W_2)P}||_2 + ||\frac{PV_2 - QW_2}{(W_1 + W_2)P}||_2.
\end{aligned}
$$

Because of the fact (2), it is established that there exists a constant $c$ associated with $\mathcal{F}$ for which $|W_1 + W_2| \geq c\mathrm{len}(x^{m,z}) > c\mathrm{len}(x^m)$, thereby leading us to conclude that:

$$
\begin{aligned}
& ||\frac{C_1\mathrm{len}(x^m)\epsilon}{W_1 + W_2}||_2 + ||\frac{C_1\mathrm{len}(x^m)\epsilon Q}{(W_1 + W_2)P}||_2 + ||\frac{PV_2 - QW_2}{(W_1 + W_2)P}||_2 \\
\leq \ & |\frac{C_1\epsilon}{c}| + |\frac{C_2\epsilon A}{c}| + |\frac{V_2 - Q/PW_2}{c\mathrm{len}(x^m)}| \\
\leq \ & |\frac{C_1\epsilon}{c}| + |\frac{C_2\epsilon A}{c}| + |\frac{2\mathrm{len}(z)e^A A}{c\mathrm{len}(x^m)}|.
\end{aligned}
$$

So to make sure that $||\frac{(PV_1 - QW_1) + (PV_2 - QW_2)}{(W_1 + W_2)P}V||_2 \leq \delta$ for the given $\delta$, based on the assumptions and above result, we just need the $\frac{C_1\epsilon}{c} \leq \delta/(3||V||_2)$, that is $m\mathrm{len}(x) \geq \frac{3C_1 C_{\mathcal{F}'}\mathrm{len}(z)||V||_2}{\delta c}$; and

$\frac{C_2 \epsilon A}{c} \leq \delta/(3||V||_2)$, that is $m\text{len}(x) \geq \frac{3C_2 C_{\mathcal{F}'} A \text{len}(z)||V||_2}{c\delta}$; and $\frac{2\text{len}(z)e^A A}{c\text{len}(x^m)} \leq \delta/(3||V||_2)$, that is $m\text{len}(x) \geq \frac{6\text{len}(z)e^A A||V||_2}{c\delta}$.

Combining them, define $C_{\mathcal{F}}^A = \max\{3C_1 C_{\mathcal{F}'}/c, 3C_2 C_{\mathcal{F}}' A/c, 6e^A A/c\}||V||_2$, which is only dependent on $\mathcal{F}$, and we know that $C_{\mathcal{F}}^A$ satisfies that when $m\text{len}(x) \geq \frac{C_{\mathcal{F}}^A \text{len}(z)}{\epsilon}$, there is $||\text{ATT}_L(\mathcal{F}_{L-1}(x^m))[\text{len}(x^m)] - \text{ATT}_L(\mathcal{F}_{L-1,\text{len}(x^{m,z})}(x^{m,z}))[\text{len}(x^{m,z})]||_2 \leq \epsilon$. So we prove the result in this part.

**We now prove the lemma.** It is easy to see that $||\text{FNN}_L(x) - \text{FNN}_L(z)||_2 \leq ||W_L||_2 ||x - z||_2$ for any $x$ and $z$, where $W_L$ are the parameters in $\text{FNN}_L$ layer, so we have:

$$
\begin{aligned}
&||\mathcal{F}_{L,\text{len}(x^m)}(x^m) - \mathcal{F}_{L,\text{len}(x^{m,z})}(x^{m,z})||_2 \\
\leq\ &(||W_L||_2 + 1)(||\text{ATT}_L(\mathcal{F}_{L-1}(x^m))[\text{len}(x^m)] \\
&- \text{ATT}_L(\mathcal{F}_{L-1,\text{len}(x^{m,z})}(x^{m,z}))[\text{len}(x^{m,z})]||_2 \\
&+ ||\mathcal{F}_{L-1,\text{len}(x^m)}(x^m) - \mathcal{F}_{L-1,\text{len}(x^{m,z})}(x^{m,z})||_2).
\end{aligned}
$$

So to make sure that $||\mathcal{F}_{L,\text{len}(x^m)}(x^m) - \mathcal{F}_{L,\text{len}(x^{m,z})}(x^{m,z})||_2 \leq \epsilon$ for a given $\epsilon$, we just need the $||\text{ATT}_L(\mathcal{F}_{L-1}(x^m))[\text{len}(x^m)] - \text{ATT}_L(\mathcal{F}_{L-1}(x^{m,z}))[\text{len}(x^{m,z})]||_2 \leq \frac{\epsilon}{2(1+||W_L||_2)}$ and $||\mathcal{F}_{L-1,\text{len}(x^m)}(x^m) - \mathcal{F}_{L-1,\text{len}(x^{m,z})}(x^{m,z})||_2 \leq \frac{\epsilon}{2(1+||W_L||_2)}$. By the above part and assumptions, we just need to take $C_{\mathcal{F}} = 2(1 + ||W_L||_2) \max\{C_{\mathcal{F}}^A, C_{\mathcal{F}'}\}$, then when $m\text{len}(x) \geq C_{\mathcal{F}}\text{len}(z)/x$, there are $||\mathcal{F}_{L,\text{len}(x^m)}(x^m) - \mathcal{F}_{L,\text{len}(x^{m,z})}(x^{m,z})||_2 \leq \epsilon$, such $C_{\mathcal{F}}$ is only dependent on $\mathcal{F}$, so we prove the lemma. $\qquad\square$

Then we can prove Proposition 4.8.

*Proof.* Let $\text{Con}_{\mathcal{F}}(x) = \mathcal{F}_{\max}(x) - \mathcal{F}_{\max,2}(x)$, where $\mathcal{F}_{\max}(x)$ is the max weight of $\mathcal{F}(x)$ and $\mathcal{F}_{max,2}(x)$ is the second largest weight of $\mathcal{F}(x)$.

For any $\gamma_s, \gamma_t \in \Gamma$, let $T_{\gamma_s,\gamma_t} = \text{argmin}_{x \in 2^\Gamma, x[\text{len}(x)]=\gamma_s, \widehat{\mathcal{F}}_{-M}(x)=\gamma_t}\{\text{len}(x)\}$ and $U_{\gamma_s,\gamma_t} = \min_{x \in T_{\gamma_s,\gamma_t}}\{\text{Con}_{\mathcal{F}_{-M}}(x)\}$. Because $T_{\gamma_s,\gamma_t}$ is a finite set, so it is easy to see that there is a $U_m, U_l \in \mathbb{Z}_+$ such that: for any $s, t$ and $x \in T_{\gamma_s,\gamma_t}$, there is $\widehat{\mathcal{F}}(x^m) = \widehat{\mathcal{F}}_{-M}(x^m)$ for any $m \geq U_m$; for any $s, t$ and $x \in T_{\gamma_s,\gamma_t}$, there is $\text{len}(x) \leq U_l$.

Then for a $\mathcal{F}$-lead set $S$ and a $(x,y) \in S$, for a $z \in T_{x[\text{len}(x)],y}$ such that $z[\text{len}(z)] = x[\text{len}(x)]$ and $y = \widehat{\mathcal{F}}_{-M}(z)$, by Lemma E.1, there is a $C$ and when we take a prompt $z^m$ such that $\text{len}(z^m) \geq \max\{C\text{len}(x)/U_{x[\text{len}(x)],y}, \text{len}(z)U_m\}$, there are $\widehat{\mathcal{F}}(z^m \oplus x) = \widehat{\mathcal{F}}(z^m) = \widehat{\mathcal{F}}_{-M}(z^m) = \widehat{\mathcal{F}}_{-M}(z) = y$. So to make sure $\widehat{\mathcal{F}}(z^m \oplus x) = y$ for any $(x,y) \in S$, we just need to make sure that $\text{len}(z^m) \geq \{C\max_{(x,y)\in S}\{\text{len}(x)/U_{x[\text{len}(x)],y}\}, U_l U_m\}$, use $\text{len}(z) \leq U_l$ here.

Finally, since $U_l, U_m$ only depend on $\mathcal{F}$, we take $U_{\mathcal{F}} = \max\{C/\min_{i,j,T_{\gamma_i,\gamma_j}\neq\phi}\{U_{\gamma_i,\gamma_j}\}, U_l U_m\}$ which can lead to the result. $\qquad\square$

### E.2 Proof of Proposition 4.9

*Proof.* We consider a one-hidden layer transformer $\mathcal{F}$ as follows:

(1) Embedding layer: embedding $\gamma_2$ to $(0.5, 0.5)$, and $\gamma_i \in \Gamma/\gamma_2$ to $v_i$ such that: $\sum v_i = -1$, where $\sum v$ is the sum of each component of vector $v$.

(2) Attention layer, let $QK = 0$ and $V = 2I$. So the last row of the attention layer is calculated as $\text{ATT}(x)[\text{len}(x)] = v_{x[\text{len}(x)]} + 2\frac{\sum_{j=1}^{\text{len}(x)} v_{x_j}}{\text{len}(x)}$, where $x_j$ is the $j$-th symbol in $x$, and $v_{x_j}$ is the embedding vector of $x_j$.

(3) The output layer and the FNN layer, just let them be $\mathcal{F}(x) = W(\text{Relu}(v_{x[\text{len}(x)]} + \text{ATT}(x)[\text{len}(x)]) + v_{x[\text{len}(x)]} + \text{ATT}(x)[\text{len}(x)])$, where $W \in \mathbb{R}^{|\Gamma| \times 2}$, and the second row of $W$ is $(1,1)$, the other rows are all negative, $v_x$ is the embedding matrix of $x$.

Let $(\cdot)_i$ mean the $i$-th row of the given matrix. It is easy to see that, if and only if $\sum(v_{x[\text{len}(x)]} + \text{ATT}(x)[\text{len}(x)]) > 0$, $\mathcal{F}(x)$ has label $\gamma_2$. Let $2^\Gamma_{\neq 2} = \{(\gamma_{i_1}, \gamma_{i_2}, \gamma_{i_3}, \ldots, \gamma_{i_m}) \| i_j \neq 2\} \subset 2^\Gamma$.

Then we consider the following $PPP$ question: $\mathcal{F}$ is given above, $S = \{(x_m, \gamma_2) \| x_m \in 2^\Gamma_{\neq 2}, x_m[\text{len}(x_m)] = \gamma_1\}$, we show this is what we want.

**Part One: $S$ is $\mathcal{F}$-lead.** Because $\widehat{\mathcal{F}}_{-M}(\gamma_2^5 \oplus \gamma_1) = \gamma_2$, so such $S$ is a $\mathcal{F}$-lead.

**Part Two: Prove the result.**

For any given $x \in 2^\Gamma_{\neq 2}$ and let $\text{len}(x) = m$. If we want to use prompt $Pr$ to make $\widehat{\mathcal{F}}(\mathcal{P} \oplus x) = \gamma_2$, write $x_p = \mathcal{P} \oplus x$, based on the definition of the output layer and the FNN layer, we need to make sure that $\sum(v_{x_p[\text{len}(x_p)]} + \text{ATT}(x_p)[\text{len}(x_p)]) > 0$, which implies that: $-1 + 2\sum_{j:(x_p)_j=\gamma_2} 1/\text{len}(x_p) - 2\sum_{j:(x_p)_j \neq \gamma_2} 1/\text{len}(x_p) > 0$, so that: $-1 + 2\frac{2\text{Num}_{\gamma_2}(x_p) - (\text{len}(\mathcal{P})+m)}{\text{len}(\mathcal{P})+m} > 0$, that is: $\text{Num}_{\gamma_2}(x_p) > 3/4(\text{len}(\mathcal{P}) + m)$, consider that $\text{Num}_{\gamma_2}(x_p) = \text{Num}_{\gamma_2}(\mathcal{P})$, so there must be $\text{len}(\mathcal{P}) > 3\text{len}(x) = 3m$.

So, for such a dataset, if $\mathcal{P}$ makes $\text{PPP}(\Gamma, S, \mathcal{F}) = 1$, by the above result, there are $\text{len}(\mathcal{P}) > 3\max_{(x,y)\in S}\{\text{len}(x)\}$, and this is what we want. $\qquad\square$

# F PROOFS OF SECTION 4.4

## F.1 PROOF OF PROPOSITION 4.13

*Proof.* Without loss of generality, for a symbol set $\Gamma = \{\gamma_i\}_{i=1}^m$ where $m \geq 3$ and a data set $S = \{(x_1, \gamma_1), (x_2, \gamma_2), \ldots\}$, where $\text{last}(x_1) = \text{last}(x_2) = \gamma_k$ and $k \in [m]$, we will show that there is a $\mathcal{F}$ satisfied condition (3) such that if prompt $\mathcal{P}$ satisfied $\widehat{\mathcal{F}}(\mathcal{P} \oplus x_1) = \gamma_1$, then $\widehat{\mathcal{F}}(\mathcal{P} \oplus x_2) \neq \gamma_2$, which can directly prove the proposition.

Let $\text{len}(S) = L$, and $M \in \mathbb{Z}_+$ such that $M > 3(L + m)$, let $p \in [m]$ such that $p \neq k$. We will consider the transformer $\mathcal{F}$ with one hidden layer as follows:

(1) Embedding layer: the embedding layer of $\gamma_i$ is $v_i$, and satisfies that $v_i \perp v_j$ and $\|v_i\|_2 = 1$ for any $i \neq j$;

(2) Hidden layer: In the attention layer, $QK = 0$ and $V = I$; the FNN layer is 0; so the last row of the hidden layer $\mathcal{F}_l(x)$ is calculated as $\mathcal{F}_l(x) = v_{x_{\text{len}(x)}} + \frac{\sum_{j=1}^{\text{len}(x)} v_{x_j}}{\text{len}(x)}$, where $x_j$ is the $j$-th symbol in $x$, and $v_{x_j}$ is the embedding vector of $x_j$

(3) Output layer and FNN layer: The output layer of $\mathcal{F}$ is that: the first weight is $\mathcal{F}_1(x) = 2M(m+1)^m(v_p\mathcal{F}_l(x)^T - 1 + 1/M)$, the $i$-th weight is $\mathcal{F}_i(x) = (m+1)^{i-2}(v_p(\mathcal{F}_l(x))^T - (i-1)/i + 1/(m+1))$, where $i \geq 2$.

Easy to check that $\widehat{\mathcal{F}}(\gamma_p^{i-1} \oplus \gamma_k) = \gamma_i$ for $i \geq 2$ and $\widehat{\mathcal{F}}(\gamma_p^{2M} \oplus \gamma_k) = \gamma_1$, so based on the conditions (2), we know that the conditions (3) is satisfied for such $\mathcal{F}$ and $S$.

Now we show that: if prompt $\mathcal{P}$ satisfied $\widehat{\mathcal{F}}(\mathcal{P} \oplus x_1) = \gamma_1$, then $\widehat{\mathcal{F}}(\mathcal{P} \oplus x_2) \neq \gamma_2$.

**Part One:** To make sure that $\widehat{\mathcal{F}}(\mathcal{P} \oplus x_1) = \gamma_1$, there must be $-1 + 1/M + v_p(\mathcal{F}_l(x))^T > 0$. If not, to make sure that $\mathcal{F}_1$ is large than $\mathcal{F}_i$ when $i \geq 2$, there must be $0 > \mathcal{F}_1 > \mathcal{F}_2$, which implies $v_p(\mathcal{F}_l(x))^T < 0.5 - 1/(m+1)$, hence, we have that $\mathcal{F}_2(x) \geq -0.5 + 1/(m+1) > -2Mm^m/3 > 2Mm^m(0.5 - 1/(m+1) - 1 + 1/M) > \mathcal{F}_1(x)$, which is contradictory to $\widehat{\mathcal{F}}(\mathcal{P} \oplus x_1) = \gamma_1$.

**Part Two:** Consider that there are $v_p(\mathcal{F}_l(x_1))^T = v_p(v_k + \frac{\sum_{j=1}^{\text{len}(x_1)+\text{len}(\mathcal{P})} v_{(\mathcal{P}\oplus x_1)_j}}{\text{len}(\mathcal{P})+\text{len}(x_1)})^T = \frac{\text{Num}_{\gamma_p}(\mathcal{P}\oplus x_1)}{\text{len}(\mathcal{P})+\text{len}(x_1)}$, by Part one, so we know that $\frac{\text{Num}_{\gamma_p}(\mathcal{P}\oplus x_1)}{\text{len}(\mathcal{P})+\text{len}(x_1)} > 1 - 1/M$.

Consider that $\frac{\text{Num}_{\gamma_p}(\mathcal{P}\oplus x_1)}{\text{len}(\mathcal{P})+\text{len}(x_1)} \leq \frac{\text{Num}_{\gamma_p}(\mathcal{P})+L-1}{\text{len}(\mathcal{P})+L} \leq 1 - \frac{1}{L+\text{len}(\mathcal{P})}$, so we have that $L + \text{len}(\mathcal{P}) \geq M$.

**Part Three :** Now we prove the result. Based on the result in Part two, we have that:

$$
\begin{aligned}
&\frac{\text{Num}_{\gamma_p}(\mathcal{P} \oplus x_2)}{\text{len}(\mathcal{P}) + \text{len}(x_2)} \\
\geq\ &\frac{\text{Num}_{\gamma_p}(\mathcal{P})}{\text{len}(\mathcal{P}) + L} \\
=\ &\frac{\text{Num}_{\gamma_p}(\mathcal{P}) + L - 1}{\text{len}(\mathcal{P}) + L} - \frac{L-1}{\text{len}(\mathcal{P}) + L} \\
\geq\ &\frac{\text{Num}_{\gamma_p}(\mathcal{P} \oplus x_1)}{\text{len}(\mathcal{P}) + \text{len}(x_1)} - \frac{L-1}{M} \\
\geq\ &1 - L/M > 2/3.
\end{aligned}
$$

Take it into transformer $\mathcal{F}$, we have that

$$
\begin{aligned}
&\mathcal{F}_2(x_2) - \mathcal{F}_3(x_2) \\
=\ &(v_p(\mathcal{F}_l(x_2))^T - 1/2 + 1/(T+1)) - (T+1)(v_p(\mathcal{F}_l(x_2))^T - 2/3 + 1/(T+1)) \\
=\ &(\frac{\text{Num}_{\gamma_p}(\mathcal{P} \oplus x_2)}{\text{len}(\mathcal{P}) + \text{len}(x_2)} - 1/2 + 1/(T+1)) - (T+1)(\frac{\text{Num}_{\gamma_p}(\mathcal{P} \oplus x_2)}{\text{len}(\mathcal{P}) + \text{len}(x_2)} - 2/3 + 1/(T+1)) \\
\leq\ &(2/3 - 1/2 + 1/(T+1)) - (T+1)(2/3 - 2/3 + 1/(T+1)) \\
\leq\ &1/6 + 1/(T+1) - 1 < 0.
\end{aligned}
$$

So $\mathcal{F}(\mathcal{P} \oplus x_2)$ will not have label $\gamma_2$, so we prove the result. $\qquad\square$

### F.2 PROOF OF PROPOSITION 4.14

*Proof.* Without loss of generality, for a symbol set $\Gamma = \{\gamma_i\}_{i=1}^m$ where $m \geq 3$ and a data set $S = \{(x_1, \gamma_1), (x_2, \gamma_1), \dots\}$, where $\text{last}(x_1) = \gamma_k, \text{last}(x_2) = \gamma_p$ and $k \neq p \in [m]$. We show that there is a $\mathcal{F}$ satisfied the conditions (3) such that if prompt $\mathcal{P}$ satisfied $\widehat{\mathcal{F}}(\mathcal{P} \oplus x_1) = \gamma_1$, then $\widehat{\mathcal{F}}(\mathcal{P} \oplus x_2) \neq \gamma_1$, this can directly prove the proposition.

Let $\text{len}(S) = L$, and $M \in \mathbb{Z}_+$ such that $M > 2(L + m)$. We will consider the transformer $\mathcal{F}$ with one hidden layer as follows:

(1) Embedding layer: the embedding layer of $\gamma_i$ is $v_i$, and satisfies that $v_i \perp v_j$ and $||v_i||_2 = 1$ for any $i \neq j$;

(2)Hidden layer: In the attention layer, $QK = 0$ and $V = I$; the FNN layer is 0; so the last row of the hidden layer $\mathcal{F}_l(x)$ is calculated as $\mathcal{F}_l(x) = v_{x_{\text{len}(x)}} + \frac{\sum_{j=1}^{\text{len}(x)} v_{x_j}}{\text{len}(x)}$, where $x_j$ is the $j$-th symbol in $x$, and $v_{x_j}$ is the embedding vector of $x_j$

(3) Output layer and FNN layer: The output layer of $\mathcal{F}$ is that: the first weight is $\mathcal{F}_1(x) = v_p\mathcal{F}_l(x)^T - 1 + 1/M$, the 2-th weight is $\mathcal{F}_2(x) = M(v_p\mathcal{F}_l(x)^T - 1 - 1/M)$, the $i$-th weight is $\mathcal{F}_i(x) = -v_p\mathcal{F}_l(x)^T + 1 - 1/M$, where $i > 2$.

It is easy to check that $\widehat{\mathcal{F}}(\gamma_p^M \oplus \gamma_i) = \gamma_1$ for $i \neq p$ and $\widehat{\mathcal{F}}(\gamma_k^M \oplus \gamma_p) = \gamma_1$, so based on the conditions (2), we know that the conditions (3) are satisfied for such $\mathcal{F}$ and $S$.

Now we show that: if prompt $\mathcal{P}$ satisfied $\widehat{\mathcal{F}}(\mathcal{P} \oplus x_1) = \gamma_1$, then $\widehat{\mathcal{F}}(\mathcal{P} \oplus x_2) \neq \gamma_1$.

**Part One:** To make sure that $\widehat{\mathcal{F}}(\mathcal{P} \oplus x_1) = \gamma_1$, there must be $\mathcal{F}_1(\mathcal{P} \oplus x_1) > \mathcal{F}_3(\mathcal{P} \oplus x_1)$, so that $v_p\mathcal{F}_l(\mathcal{P} \oplus x_1)^T - 1 + 1/M > 0$, so that $v_p\mathcal{F}_l(\mathcal{P} \oplus x_1) = v_p(v_k + \frac{\sum_{j=1}^{\text{len}(\mathcal{P} \oplus x_1)} v_{(\mathcal{P} \oplus x_1)_j}}{\text{len}(\mathcal{P} \oplus x_1)})^T = \frac{\text{Num}_{\gamma_p}(\mathcal{P} \oplus x_1)}{\text{len}(\mathcal{P}) + \text{len}(x_1)} \geq 1 - 1/M$.

Consider that $\frac{\text{Num}_{\gamma_p}(\mathcal{P} \oplus x_1)}{\text{len}(\mathcal{P}) + \text{len}(x_1)} \leq \frac{\text{Num}_{\gamma_p}(\mathcal{P}) + L - 1}{\text{len}(\mathcal{P}) + L} \leq 1 - \frac{1}{L + \text{len}(\mathcal{P})}$, so we have that $L + \text{len}(\mathcal{P}) \geq M$.

**Part Two:** Now we prove the result. We have that:

$$\frac{\mathrm{Num}_{\gamma_p}(\mathcal{P} \oplus x_2)}{\mathrm{len}(\mathcal{P}) + \mathrm{len}(x_2)}$$

$$\geq \frac{\mathrm{Num}_{\gamma_p}(\mathcal{P})}{\mathrm{len}(\mathcal{P}) + L}$$

$$= \frac{\mathrm{Num}_{\gamma_p}(\mathcal{P}) + L - 1}{\mathrm{len}(\mathcal{P}) + L} - \frac{L-1}{\mathrm{len}(\mathcal{P}) + L}$$

$$\geq \frac{\mathrm{Num}_{\gamma_p}(\mathcal{P} \oplus x_1)}{\mathrm{len}(\mathcal{P}) + \mathrm{len}(x_1)} - \frac{L-1}{M}$$

$$\geq 1 - L/M > 0.5.$$

If we input this into a transformer, we find that

$$\mathcal{F}_2(x_2) - \mathcal{F}_1(x_2)$$
$$= M(v_p \mathcal{F}_l(x_2)^T - 1 - 1/M) - (v_p \mathcal{F}_l(x_2)^T - 1 + 1/M)$$
$$= M(1 + \frac{\mathrm{Num}_{\gamma_p}(\mathcal{P} \oplus x_2)}{\mathrm{len}(\mathcal{P}) + \mathrm{len}(x_2)} - 1 - 1/M) - (1 + \frac{\mathrm{Num}_{\gamma_p}(\mathcal{P} \oplus x_2)}{\mathrm{len}(\mathcal{P}) + \mathrm{len}(x_2)} - 1 + 1/M)$$
$$\geq M(1.5 - 1 - 1/M) - (1.5 - 1 + 1/M) > 0.$$

So we prove the result. $\square$

# G PROOFS OF SECTION 5

## G.1 PROOF OF THEOREM 5.1

We give a classic generalization bound below, which will be used in the proof.

**Theorem G.1** (P.217 of (Mohri et al., 2018), Informal). *Let the training set $\mathcal{D}_{tr}$ be i.i.d. sampled from the data distribution $\mathcal{D}_S$ and $N = |\mathcal{D}_{tr}|$. For the hypothesis space $H = \{L(\mathcal{F}(x), y) : \mathbb{R}^n \times [m] \to [0,1]\}$ and $\delta \in \mathbb{R}_+$, with probability at least $1 - \delta$, for any $L(\mathcal{F}(x), y) \in H$, we have*

$$\mathbb{E}_{(x,y) \sim \mathcal{D}_S}[L(\mathcal{F}(x), y)] \leq \mathbb{E}_{(x,y) \in \mathcal{D}_{tr}}[L(\mathcal{F}(x), y)] + 2\mathrm{Rad}_N^{\mathcal{D}_S}(H) + \sqrt{\frac{\ln(1/\delta)}{2N}} \quad (4)$$

Now we prove the Theorem 5.1

*Proof.* Let $2_L^\Gamma$ be the set of $x \in 2^\Gamma$ satisfying $\mathrm{len}(x) \leq L$. For a given transformer $\mathcal{F}$, we define the hypothetical space $H_{\mathcal{F},L}$ of function $2^\Gamma \times \Gamma \to \{0,1\}$ as: $H_{\mathcal{F},L} = \{\mathcal{G}_\mathcal{P}(x,y) \in 2^\Gamma \times \Gamma \to \{0,1\} : \mathcal{G}_\mathcal{P}(x,y) = I(\widehat{\mathcal{F}}(\mathcal{P} \oplus x) = y), \mathcal{P} \in 2_L^\Gamma\}$. It is easy to see that $H_{\mathcal{F},L}$ contains $T^L$ functions. Consider that $\mathrm{Rad}_N^{\mathcal{D}}(H) \leq \sqrt{\frac{2 \ln |H|}{N}}$ (Mohri et al., 2018) for any infinite hypothetical space with range $[-1,1]$ for all the function in it and distribution $\mathcal{D}$, so the Rademacher complexity $\mathrm{Rad}_N^{\mathcal{D}}(H_{\mathcal{F},L}) \leq \sqrt{\frac{2L \ln T}{N}}$ for any distribution $\mathcal{D}$ of $(x, y)$.

Hence, using the Rademacher generalization bound in G.1, we prove the result. $\square$

## G.2 PROOF OF PROPOSITION 5.4

*Proof.* It suffices to consider a special transformer $\mathcal{F}$ and $\mathcal{P}$.

For the given distribution $D$, we can find a finite set $S_s \subset 2^\Gamma \times \Gamma$ such that $\mathbb{P}_{(x,y) \sim \mathcal{D}}((x,y) \in S_s) \geq 1 - \epsilon/N$. So if we select $N$ i.i.d. samples in $\mathcal{D}$, with probability $(1 - \epsilon/N)^N \geq 1 - \epsilon$, all such samples in $S_s$.

Assume that $|S_s| = m$ and $S_s = \{(x_i, y_i)\}_{i=1}^m$. Let $\mathcal{F}$ satisfy that $\widehat{\mathcal{F}}(z_1 \oplus z_2 \oplus \ldots z_m \oplus x_i) = z_i$ for each $x_i \in S_s$ and $z_j \in \Gamma$ where $j \in [m]$.

Then for any $S \sim \mathcal{D}^N$, we just consider the situation that all samples in $S$ are in $S_s$, which is stand with probability at least $1 - \epsilon$.

Firstly, we can find a $\mathcal{P} = y'_1 \oplus y'_2 \oplus \ldots y'_m$ such that $y'_i = y_i$ if and only if there is $(x_i, y_i) \in S$. We show $\mathcal{P}$ is what we want.

Then by the definition of $\mathcal{F}$, for any $i \in [m]$ such that $(x_i, y_i) \in S$, there are $\widehat{\mathcal{F}}(\mathcal{P} \oplus x_i) = \widehat{\mathcal{F}}(y'_1 \oplus y'_2 \oplus \ldots y'_m \oplus x_i) = y'_i = y_i$, then we have that $\mathcal{A}_{\mathcal{F},\mathcal{P},S} = 1$.

But for $\mathcal{A}_{\mathcal{F},\mathcal{P},\mathcal{D}}$, we have that $\mathcal{A}_{\mathcal{F},\mathcal{P},\mathcal{D}} \leq 1 - \mathbb{E}_{(x,y)\sim D}[I((x,y) \in S_s/S)]$, because if $(x_i, y_i) \notin S$, then $\widehat{\mathcal{F}}(\mathcal{P} \oplus x_i) = \widehat{\mathcal{F}}(y_1' \oplus y_2' \oplus \cdots \oplus y_m' \oplus x_i) = y_i' \neq y_i$, based on the definition of $S_s$ and assumption on $\mathcal{D}$, so there are $\mathcal{A}_{\mathcal{F},\mathcal{P},\mathcal{D}} \leq 1 - \mathbb{E}_{(x,y)\sim D}[I((x,y) \in S_s/S)] \leq \mathbb{E}_{(x,y)\sim D}[I((x,y) \in S)] + \mathbb{E}_{(x,y)\sim D}[I((x,y) \notin S_s)] \leq \epsilon + \epsilon/N \leq 2\epsilon$.

So there are $|\mathcal{A}_{\mathcal{F},\mathcal{P},\mathcal{D}} - \mathcal{A}_{\mathcal{F},\mathcal{P},S}| \geq 1 - 2\epsilon$, which is what we want. $\qquad\square$

### G.3 PROOF OF PROPOSITION 5.5

*Proof.* Firstly, let $D$ be defined on set $S_D = \{(x_i, y_i)\} \subset 2^\Gamma \times \Gamma$ and $\mathbb{P}_{(x,y)\sim D}(x = x_i) = \mathbb{P}_i$, then there is a finite set $S_s \subset S_D$ such that $\mathbb{P}_{(x,y)\sim D}(x \in S_s) \geq 1 - \epsilon/2$.

Let $n_i$ be the number of $x_i$ in $S$, so we have that:

$$
\begin{aligned}
& |\mathcal{A}_{\mathcal{F},\mathcal{P},\mathcal{D}} - \mathcal{A}_{\mathcal{F},\mathcal{P},S}| \\
= \ & |\mathbb{E}_{(x,y)\sim D}[\mathbf{I}(\widehat{\mathcal{F}}(\mathcal{P} \oplus x) = y)] - \tfrac{1}{|S|} \sum_{x \in S}(\mathbf{I}(\widehat{\mathcal{F}}(\mathcal{P} \oplus x) = y))| \\
= \ & |\sum_{(x_i,y_i)}(\mathbb{P}_i - \tfrac{n_i}{|S|})\mathbf{I}(\widehat{\mathcal{F}}(\mathcal{P} \oplus x_i) = y_i)| \\
\leq \ & \sum_{(x_i,y_i)\in S_s} |\mathbb{P}_i - \tfrac{n_i}{|S|}| \, \mathbf{I}(\widehat{\mathcal{F}}(\mathcal{P} \oplus x_i) = y_i) + \epsilon/2
\end{aligned}
$$

Based on the Hoeffding inequality, for any $x_i$, we know that with probability $1 - 2e^{-N\epsilon^2/2}$, there is $|\mathbb{P}_i - \tfrac{n_i}{|S|}| \leq \epsilon/2$.

So we take a $N$ such that $1 - \sum_{(x_i,y_i)\in S_s} 2e^{-N\epsilon^2/2} < \delta$(because $S_s$ is finite, so such $N$ must exist), then there are $|\mathcal{A}_{\mathcal{F},\mathcal{P},\mathcal{D}} - \mathcal{A}_{\mathcal{F},\mathcal{P},S}| \leq \sum_{(x_i,y_i)\in S_s} |\mathbb{P}_i - \tfrac{n_i}{|S|}| \, \mathbf{I}(\widehat{\mathcal{F}}(\mathcal{P} \oplus x_i) = y_i) + \epsilon/2 \leq \epsilon$ with probability $1 - \delta$, which is what we want. $\qquad\square$

