# OpenReview forum: "Towards a Theoretical Understanding of Prompt Engineering: Tractability, Existence, and Generalization"
_ICLR.cc/2026/Conference — ICLR 2026 Conference Withdrawn Submission_

### Official Review · Reviewer_vioE · 2025-10-17

**Soundness:** 2
**Presentation:** 2
**Contribution:** 2
**Rating:** 4
**Confidence:** 3

**Summary:**

This paper provides a foundational theoretical analysis of prompt engineering, a field largely driven by empirical methods. The work rigorously establishes the computational hardness of finding optimal prompts, a widely held but previously unproven belief, which stands as a significant strength and contribution. The paper is well-structured around the core questions of tractability, existence, and generalization, offering valuable theoretical insights.

**Strengths:**

(1) The paper establishes the first formal proofs showing that determining whether a perfect prompt exists is NP-complete, and that optimizing a prompt is NP-hard. These results represent a major theoretical advance, offering a rigorous foundation for why prompt optimization is inherently difficult.

(2) The paper is notably well-structured, organized around three fundamental questions presented early in the introduction. By informally stating the main theorems before presenting their formal versions, the authors make the central insights more approachable and comprehensible to a broad audience.

**Weaknesses:**

(1) The theoretical framework is developed exclusively for tasks with single-token outputs. While the authors acknowledge this limitation, it remains a significant constraint, as many key applications of LLMs involve generating complex, multi-token sequences. The paper does not address the theoretical difficulties that would arise when extending the framework to sequence generation.

(2) The experiments rely on GPT-2 and RoBERTa, which no longer represent the current state of the art in LLM research. Furthermore, the datasets are synthetically constructed to satisfy the paper’s theoretical assumptions (e.g., ensuring that all sentences share the same final symbol), rather than drawn from established NLP benchmarks. As a result, the empirical results provide limited insight into the practical relevance of the theoretical findings.

(3) The sufficient condition for prompt existence (the “F-lead” dataset condition in Definition 4.3) depends on the behavior of a non-autoregressive version of the transformer. Since this object is purely theoretical and cannot be instantiated in practice, the condition is effectively unverifiable for any real-world dataset or model.

**Questions:**

Please refer to weaknesses.

---

### Official Review · Reviewer_exWa · 2025-10-26

**Soundness:** 3
**Presentation:** 2
**Contribution:** 2
**Rating:** 4
**Confidence:** 3

**Summary:**

The paper studies prompt engineering from a theoretical perspective.  The results include showing the complexity of determining the existence of an optimal prompt, separate sufficient and necessary conditions for the same, and a generalization bound.

**Strengths:**

1. Attempts to provide a theoretical foundation to a popular applied, engineering problem, which is appreciated.
2. The theory seems to be done correctly and is presented reasonably well.
3. Some results have provide neat practical guidelines: such as on the length of the prompt to be of the same order as that of the queries (Remark 4.11).

**Weaknesses:**

See questions.

My gripe is that the setting (single-token output)  is too simplified and the results not interesting/surprising/insightful enough.  As such, I cannot recommend acceptance.  I will be happy to improve my score if the authors can convince me that there are insights that I am missing.

**Questions:**

1. The paper assumes the existence of a single-token "answer" to each query.  Such an "answer" might not exist at all, and it certainly need not be a single token.  Use-cases involving the existence of a correct answer likely have a well-formulated question, making the "prompt" inessential, is my belief.  Comments?
2. The results also seem rather bizarre... Theorem 1.2/4.4 says that if all answers in the dataset are the same and all queries end with the same last token, then there exists an optimal prompt, but isn't this obvious: the answer is a "constant", so the prompt could just tell the LLM to ignore the query and output the constant answer?
3. Corollary 4.10 also just seems incorrect: as stated, it means that there exists a transformer such that if Len(P) = const Len(S), then P is a perfect prepended prompt for S.  This, of course, cannot be true --- I can come up with any number of irrelevant, gibberish prompts with the same length.

Minor:
1. Proposition 5.4 and the text following it has a "kappa" --- I believe this should be a Gamma.
2. Section 6: the paragraph titles "same target label" and "same last symbol" seem to be interchanged.

---

### Official Review · Reviewer_AfAq · 2025-10-30

**Soundness:** 3
**Presentation:** 4
**Contribution:** 2
**Rating:** 4
**Confidence:** 3

**Summary:**

The paper studies the theoretical foundations of prepended prompt engineering for autoregressive transformers. It (i) proves that deciding the existence of a perfect prompt is NP-complete and that discrete prompt optimization is NP-hard; (ii) gives sufficient structural conditions under which perfect prompts exist; (iii) derives a uniform generalization bound that scales with prompt length and sample size. The paper also reports experiments to illustrate the theories.

**Strengths:**

S1 (interesting topics): The paper targets three clearly motivated questions in prompt engineering: tractability, existence, and generalization.

S2 (complexity characterizations): The paper shows that deciding the existence of a perfect prompt is NP-complete and that discrete prompt optimization is NP-hard.

S3 (empirical illustration): The paper reports experiments to illustrate the theories.

S4 (nice presentation): This paper is well written and easy to follow.

**Weaknesses:**

W1 (technical depth): Some of the theoretical results lack technical depth a little bit. For instance, the learnability result Theorem 5.1 seems to be a direct combination of the classic Rademacher complexity bound and the Rademacher complexity of Transformers by Mohri et al. (2018), as discussed in the paper.

W2 (insufficient justifications): Some claims need more careful justifications. For example, Remark 4.11 claims that longer prompts are needed for longer sentences, but [1] proves that there exists a Transformer on which constant-length prompts suffice for arbitrarily long sentences. In this regard, the implication of Proposition 4.9 might be a bit misleading for practioners.

W3 (restrictive assumption): The "F-lead" assumption seems to be a bit restrictive and might lack practical implications. The paper did not discuss when this assumption holds in practice.

W4 (missing related work): Some related works on prompt existence are not discussed. For example, this paper only shows prompt existence under a very restrictive setting while [1] shows that there exists a Transformer on which prompting is Turing-complete.

- [1] Qiu et al. On the Turing completeness of prompting. ICLR 2025.

**Questions:**

See weaknesses. I would like to raise my rating if my concerns are addressed.

---

### Official Review · Reviewer_GP8L · 2025-10-31

**Soundness:** 2
**Presentation:** 3
**Contribution:** 2
**Rating:** 2
**Confidence:** 3

**Summary:**

In this paper, the authors study the problem of prompt engineering from a theoretical perspective. In particular, they study three problems: (1) the existence of perfect prompts; (2) the computability of perfect prompts; (3) the generalizability of prompts to the whole data distribution. Experiments are carried out to validate the theoretical findings.

**Strengths:**

The paper tackles an important and timely problem. Theoretical results on this topic are scarce, so I appreciate the authors' efforts in closing this gap. The paper's presentation is overall good.

**Weaknesses:**

My main concern about the paper is that I feel that the problem of finding perfect prompts is unnecessarily hard. A prompt that works perfectly for the entire dataset seems too much to ask, especially given that this result would then be used in combination to Theorem 5.1, which is a high-probability result with vanishing error. A more sensible setting would be to relax the assumptions and only ask for approximately perfect prompts, that work for most of the data points.

Probably due to the very strict constraints, the results presented in the paper feel quite weak to me. Theorem 4.1 is interesting. Even though it is hardly surprising, providing a formal proof is still valuable (although I would like some clarifications from the authors about its proof, see below). On the contrary, Theorem 4.4 feels very weak as it involves a dataset where all queries in the dataset have the very same answer. Since this should then be applied in conjunction with Theorem 5.1, this would lead to the conclusion that all queries should have the exact same answer with high probability, which feels way too impractical. Theorems in Sec. 4.4 also play around the same strict requirements of having the same answer to all queries in the dataset. Furthermore, all these results seem to rely heavily on the absence of positional encoding in the architecture, another very limiting assumption (Lemma B.2), which also seems violated in the experiments, which involve GPT and RoBERTa. Theorem 5.1 is actually useful, but it is a simple application of previous results (Mohri et al., 2018).

**Questions:**

Mainly, I would like to ask the authors if they could defend the strong assumptions on which their results are based. Specifically:
1. Is it correct that most of the results require the absence of positional embeddings? How would they change if those are included?
2. Can you provide some practical justification for the assumption that all queries in S should have the very same answer in Theorem 4.4?
3. Regarding the proof in Theorem 4.1, I am a bit confused about the fact that you are free to design the transformer and the dataset. I thought that they would be given. Can you explain to me the reasoning behind the proof?
4. Can you explain why in Proposition 5.4, the condition on the max means that the distribution is concentrated on N points? Superficially, it seems that requiring that the maximum probability is small seems to imply exactly the opposite.

---

### Author Response · Authors · 2025-11-14

The reviewers’ main concern is that we oversimplified the model. We thank them for pointing out this limitation and acknowledge it as a weakness that we will address in the revised version.

---

### Note · Authors · 2025-11-14

I have read and agree with the venue's withdrawal policy on behalf of myself and my co-authors.